# Multiple Source Domain Adaptation with Adversarial Learning

## Abstract

While domain adaptation has been actively researched in recent years, most theoretical results and algorithms focus on the single-source-single-target adaptation setting. Naive application of such algorithms on multiple source domain adaptation problem may lead to suboptimal solutions. We propose a new generalization bound for domain adaptation when there are multiple source domains with labeled instances and one target domain with unlabeled instances. Compared with existing bounds, the new bound does not require expert knowledge about the target distribution, nor the optimal combination rule for multisource domains. Interestingly, our theory also leads to an efficient learning strategy using adversarial neural networks: we show how to interpret it as learning feature representations that are invariant to the multiple domain shifts while still being discriminative for the learning task. To this end, we propose two models, both of which we call multisource domain adversarial networks (MDANs): the first model optimizes directly our bound, while the second model is a smoothed approximation of the first one, leading to a more data-efficient and task-adaptive model. The optimization tasks of both models are minimax saddle point problems that can be optimized by adversarial training. To demonstrate the effectiveness of MDANs, we conduct extensive experiments showing superior adaptation performance on three real-world datasets: sentiment analysis, digit classification, and vehicle counting.

## 1 Introduction

The success of machine learning algorithms has been partially attributed to rich datasets with abundant annotations (Krizhevsky et al., 2012; Hinton et al., 2012; Russakovsky et al., 2015). Unfortunately, collecting and annotating such large-scale training data is prohibitively expensive and time-consuming. To solve these limitations, different labeled datasets can be combined to build a larger one, or synthetic training data can be generated with explicit yet inexpensive annotations (Shrivastava et al., 2016). However, due to the possible shift between training and test samples, learning algorithms based on these cheaper datasets still suffer from high generalization error. Domain adaptation (DA) focuses on such problems by establishing knowledge transfer from a labeled source domain to an unlabeled target domain, and by exploring domain-invariant structures and representations to bridge the gap (Pan & Yang, 2010). Both theoretical results (Ben-David et al., 2010; Mansour et al., 2009a; Mansour & Schain, 2012; Xu & Mannor, 2012; Gopalan et al., 2014) and algorithms (Becker et al., 2013; Hoffman et al., 2012; Ajakan et al., 2014; Ghifary et al., 2015; Jhuo et al., 2012) for DA have been proposed. Recently, DA algorithms based on deep neural networks produce breakthrough performance by learning more transferable features (Glorot et al., 2011; Donahue et al., 2014; Yosinski et al., 2014; Bousmalis et al., 2016; Long et al., 2015). Most theoretical results and algorithms with respect to DA focus on the single-source-single-target adaptation setting (Ganin et al., 2016; Tzeng et al., 2015; 2017). However, in many application scenarios, the labeled data available may come from multiple domains with different distributions. As a result, naive application of the single-source-single-target DA algorithms may lead to suboptimal solutions. Such problem calls for an efficient technique for multiple source domain adaptation. Hoffman et al. (2012); Gan et al. (2016); Zhang et al. (2015) explore multisource DA methods, but they are based on non-deep architectures and their performance have much space to be improved.

In this paper, we theoretically analyze the multiple source domain adaptation problem and propose an adversarial learning strategy based on our theoretical results. Specifically, we prove a new

generalization bound for domain adaptation when there are multiple source domains with labeled instances and one target domain with unlabeled instances. Our theoretical results build on the seminal theoretical model for domain adaptation introduced by Ben-David et al. (2010), where a divergence measure, known as the $\mathcal{H}$-divergence, was proposed to measure the distance between two distributions based on a given hypothesis space $\mathcal{H}$. Our new result generalizes the bound (Ben-David et al., 2010, Thm. 2) to the case when there are multiple source domains. The new bound has an interesting interpretation and reduces to (Ben-David et al., 2010, Thm. 2) when there is only one source domain. Technically, we derive our bound by first proposing a generalized $\mathcal{H}$-divergence measure between two sets of distributions from multi-domains. We then prove a PAC bound (Valiant, 1984) for the target risk by bounding it from empirical source risks, using tools from concentration inequalities and the VC theory (Vapnik, 1998). Compared with existing bounds, the new bound does not require expert knowledge about the target domain distribution (Mansour et al., 2009b), nor the optimal combination rule for multiple source domains (Ben-David et al., 2010). Our results also imply that it is not always beneficial to naively incorporate more source domains into training, which we verify to be true in our experiments.

Interestingly, our bound also leads to an efficient implementation using adversarial neural networks. This implementation learns both domain invariant and task discriminative feature representations under multiple domains. Specifically, we propose two models (both named MDANs) by using neural networks as rich function approximators to instantiate the generalization bound we derive (Fig. 1). After proper transformations, both models can be viewed as computationally efficient approximations of our generalization bound, so that the goal is to optimize the parameters of the networks in order to minimize the bound. The first model optimizes directly our generalization bound, while the second is a smoothed approximation of the first, leading to a more data-efficient and task-adaptive model. The optimization problem for each model is a minimax saddle point problem, which can be interpreted as a zero-sum game with two participants competing against each other to learn invariant features. Both models combine feature extraction, domain classification, and task learning in one training process. MDANs is generalization of the popular domain adversarial neural network (DANN) (Ganin et al., 2016) and reduce to it when there is only one source domain. We propose to use stochastic optimization with simultaneous updates to optimize the parameters in each iteration. To demonstrate the effectiveness of MDANs as well as the relevance of our theoretical results, we conduct extensive experiments on real-world datasets, including both natural language and vision tasks. We achieve superior adaptation performances on all the tasks, validating the effectiveness of our models.

## 2 PRELIMINARY

We first introduce the notation used in this paper and review a theoretical model for domain adaptation when there is only one source and one target domain (Kifer et al., 2004; Ben-David et al., 2007; Blitzer et al., 2008; Ben-David et al., 2010). The key idea is the $\mathcal{H}$-divergence to measure the discrepancy between two distributions. Other theoretical models for DA exist (Cortes et al., 2008; Mansour et al., 2009a;c; Cortes & Mohri, 2014); we choose to work with the above model because this distance measure has a particularly natural interpretation and can be well approximated using samples from both domains.

**Notations**  We use *domain* to represent a distribution $\mathcal{D}$ on input space $\mathcal{X}$ and a labeling function $f : \mathcal{X} \to [0, 1]$. In the setting of one source one target domain adaptation, we use $\langle \mathcal{D}_S, f_S \rangle$ and $\langle \mathcal{D}_T, f_T \rangle$ to denote the source and target domain, respectively. A *hypothesis* is a binary classification function $h : \mathcal{X} \to \{0, 1\}$. The *error* of a hypothesis $h$ w.r.t. a labeling function $f$ under distribution $\mathcal{D}_S$ is defined as: $\varepsilon_S(h, f) := \mathbb{E}_{\mathbf{x} \sim \mathcal{D}_S}[|h(\mathbf{x}) - f(\mathbf{x})|]$. When $f$ is also a hypothesis, then this definition reduces to the probability that $h$ disagrees with $f$ under $\mathcal{D}_S$: $\mathbb{E}_{\mathbf{x} \sim \mathcal{D}_S}[|h(\mathbf{x}) - f(\mathbf{x})|] = \mathbb{E}_{\mathbf{x} \sim \mathcal{D}_S}[\mathbb{I}(f(\mathbf{x}) \neq h(\mathbf{x}))] = \Pr_{\mathbf{x} \sim \mathcal{D}_S}(f(\mathbf{x}) \neq h(\mathbf{x}))$.

We define the *risk* of hypothesis $h$ as the error of $h$ w.r.t. a true labeling function under domain $\mathcal{D}_S$, i.e., $\varepsilon_S(h) := \varepsilon_S(h, f_S)$. As common notation in computational learning theory, we use $\widehat{\varepsilon}_S(h)$ to denote the empirical risk of $h$ on the source domain. Similarly, we use $\varepsilon_T(h)$ and $\widehat{\varepsilon}_T(h)$ to mean the true risk and the empirical risk on the target domain. $\mathcal{H}$-divergence is defined as follows:

**Definition 2.1.** Let $\mathcal{H}$ be a hypothesis class for instance space $\mathcal{X}$, and $\mathcal{A}_{\mathcal{H}}$ be the collection of subsets of $\mathcal{X}$ that are the support of some hypothesis in $\mathcal{H}$, i.e., $\mathcal{A}_{\mathcal{H}} := \{h^{-1}(\{1\}) \mid h \in \mathcal{H}\}$. The distance

between two distributions $\mathcal{D}$ and $\mathcal{D}'$ based on $\mathcal{H}$ is:

$$d_{\mathcal{H}}(\mathcal{D}, \mathcal{D}') := 2 \sup_{A \in \mathcal{A}_{\mathcal{H}}} |\Pr_{\mathcal{D}}(A) - \Pr_{\mathcal{D}'}(A)|$$

When the hypothesis class $\mathcal{H}$ contains all the possible measurable functions over $\mathcal{X}$, $d_{\mathcal{H}}(\mathcal{D}, \mathcal{D}')$ reduces to the familiar total variation. Given a hypothesis class $\mathcal{H}$, we define its symmetric difference w.r.t. itself as: $\mathcal{H}\Delta\mathcal{H} = \{h(\mathbf{x}) \oplus h'(\mathbf{x}) \mid h, h' \in \mathcal{H}\}$, where $\oplus$ is the xor operation. Let $h^*$ be the optimal hypothesis that achieves the minimum combined risk on both the source and the target domains:

$$h^* := \arg\min_{h \in \mathcal{H}} \varepsilon_S(h) + \varepsilon_T(h)$$

and use $\lambda$ to denote the combined risk of the optimal hypothesis $h^*$:

$$\lambda := \varepsilon_S(h^*) + \varepsilon_T(h^*)$$

Ben-David et al. (2007) and Blitzer et al. (2008) proved the following generalization bound on the target risk in terms of the source risk and the discrepancy between the source domain and the target domain:

**Theorem 2.1** ((Blitzer et al., 2008)). Let $\mathcal{H}$ be a hypothesis space of $VC$-dimension $d$ and $\mathcal{U}_S, \mathcal{U}_T$ be unlabeled samples of size $m$ each, drawn from $\mathcal{D}_S$ and $\mathcal{D}_T$, respectively. Let $\widehat{d}_{\mathcal{H}\Delta\mathcal{H}}$ be the empirical distance on $\mathcal{U}_S$ and $\mathcal{U}_T$; then with probability at least $1 - \delta$ over the choice of samples, for each $h \in \mathcal{H}$,

$$\varepsilon_T(h) \leq \varepsilon_S(h) + \frac{1}{2}\widehat{d}_{\mathcal{H}\Delta\mathcal{H}}(\mathcal{U}_S, \mathcal{U}_T) + 4\sqrt{\frac{2d\log(2m) + \log(4/\delta)}{m}} + \lambda \tag{1}$$

The generalization bound depends on $\lambda$, the optimal combined risk that can be achieved by hypothesis in $\mathcal{H}$. The intuition is that if $\lambda$ is large, then we cannot hope for a successful domain adaptation. One notable feature of this bound is that the empirical discrepancy distance between two samples $\mathcal{U}_S$ and $\mathcal{U}_T$ can usually be approximated by a discriminator to distinguish instances from these two domains.

## 3 A NEW GENERALIZATION BOUND FOR MULTIPLE SOURCE DOMAIN ADAPTATION

In this section we first generalize the definition of the discrepancy function $d_{\mathcal{H}}(\cdot, \cdot)$ that is only appropriate when we have two domains. We will then use the generalized discrepancy function to derive a generalization bound for multisource domain adaptation. We conclude this section with a discussion and comparison of our bound and existing generalization bounds for multisource domain adaptation (Mansour et al., 2009c; Ben-David et al., 2010). We refer readers to appendix for proof details and we mainly focus on discussing the interpretations and implications of the theorems.

Let $\{\mathcal{D}_{S_i}\}_{i=1}^{k}$ and $\mathcal{D}_T$ be $k$ source domains and the target domain, respectively. We define the discrepancy function $d_{\mathcal{H}}(\mathcal{D}_T; \{\mathcal{D}_{S_i}\}_{i=1}^{k})$ induced by $\mathcal{H}$ to measure the distance between $\mathcal{D}_T$ and a set of domains $\{\mathcal{D}_{S_i}\}_{i=1}^{k}$ as follows:

**Definition 3.1.**

$$d_{\mathcal{H}}(\mathcal{D}_T; \{\mathcal{D}_{S_i}\}_{i=1}^{k}) := \max_{i \in [k]} d_{\mathcal{H}}(\mathcal{D}_T; \mathcal{D}_{S_i}) = 2\max_{i \in [k]} \sup_{A \in \mathcal{A}_{\mathcal{H}}} |\Pr_{\mathcal{D}_T}(A) - \Pr_{\mathcal{D}_{S_i}}(A)|$$

Again, let $h^*$ be the optimal hypothesis that achieves the minimum combined risk:

$$h^* := \arg\min_{h \in \mathcal{H}} \left( \varepsilon_T(h) + \max_{i \in [k]} \varepsilon_{S_i}(h) \right)$$

and define

$$\lambda := \varepsilon_T(h^*) + \max_{i \in [k]} \varepsilon_{S_i}(h^*)$$

i.e., the minimum risk that is achieved by $h^*$. The following lemma holds for $\forall h \in \mathcal{H}$:

**Theorem 3.1.** $\varepsilon_T(h) \leq \max_{i \in [k]} \varepsilon_{S_i}(h) + \frac{1}{2}d_{\mathcal{H}\Delta\mathcal{H}}(\mathcal{D}_T; \{\mathcal{D}_{S_i}\}_{i=1}^{k}) + \lambda$.

**Remark**. Let us take a closer look at the generalization bound: to make it small, the discrepancy measure between the target domain and the multiple source domains need to be small. Otherwise we cannot hope for successful adaptation by only using labeled instances from the source domains. In this case there will be no hypothesis that performs well on both the source domains and the target domain. It is worth pointing out here that the second term and the third term together introduce a tradeoff (regularization) on the complexity of our hypothesis class $\mathcal{H}$. Namely, if $\mathcal{H}$ is too restricted, then the third term $\lambda$ can be large while the discrepancy term can be small. On the other hand, if $\mathcal{H}$ is very rich, then we expect the optimal error, $\lambda$, to be small, while the discrepancy measure $d_{\mathcal{H}\Delta\mathcal{H}}(\mathcal{D}_T; \{\mathcal{D}_{S_i}\}_{i=1}^k)$ to be large. The first term is a standard source risk term that usually appears in generalization bounds under the PAC-learning framework (Valiant, 1984; Vapnik, 1998). Later we shall upper bound this term by its corresponding empirical risk.

The discrepancy distance $d_{\mathcal{H}\Delta\mathcal{H}}(\mathcal{D}_T; \{\mathcal{D}_{S_i}\}_{i=1}^k)$ is usually unknown. However, we can bound $d_{\mathcal{H}\Delta\mathcal{H}}(\mathcal{D}_T; \{\mathcal{D}_{S_i}\}_{i=1}^k)$ from its empirical estimation using $i.i.d.$ samples from $\mathcal{D}_T$ and $\{\mathcal{D}_{S_i}\}_{i=1}^k$:

**Theorem 3.2.** Let $\mathcal{D}_T$ and $\{\mathcal{D}_{S_i}\}_{i=1}^k$ be the target distribution and $k$ source distributions over $\mathcal{X}$. Let $\mathcal{H}$ be a hypothesis class where $VC\dim(\mathcal{H}) = d$. If $\widehat{\mathcal{D}}_T$ and $\{\widehat{\mathcal{D}}_{S_i}\}_{i=1}^k$ are the empirical distributions of $\mathcal{D}_T$ and $\{\mathcal{D}_{S_i}\}_{i=1}^k$ generated with $m$ $i.i.d.$ samples from each domain, then for $\epsilon > 0$, we have:

$$\Pr\left(\left|d_{\mathcal{H}}(\mathcal{D}_T; \{\mathcal{D}_{S_i}\}_{i=1}^k) - d_{\mathcal{H}}(\widehat{\mathcal{D}}_T; \{\widehat{\mathcal{D}}_{S_i}\}_{i=1}^k)\right| \geq \epsilon\right) \leq 4k\left(\frac{em}{d}\right)^d \exp\left(-m\epsilon^2/8\right)$$

The main idea of the proof is to use VC theory (Vapnik, 1998) to reduce the infinite hypothesis space to a finite space when acting on finite samples. The theorem then follows from standard union bound and concentration inequalities. Equivalently, the following corollary holds:

**Corollary 3.1.** Let $\mathcal{D}_T$ and $\{\mathcal{D}_{S_i}\}_{i=1}^k$ be the target distribution and $k$ source distributions over $\mathcal{X}$. Let $\mathcal{H}$ be a hypothesis class where $VC\dim(\mathcal{H}) = d$. If $\widehat{\mathcal{D}}_T$ and $\{\widehat{\mathcal{D}}_{S_i}\}_{i=1}^k$ are the empirical distributions of $\mathcal{D}_T$ and $\{\mathcal{D}_{S_i}\}_{i=1}^k$ generated with $m$ $i.i.d.$ samples from each domain, then, for $0 < \delta < 1$, with probability at least $1 - \delta$ (over the choice of samples), we have:

$$\left|d_{\mathcal{H}}(\mathcal{D}_T; \{\mathcal{D}_{S_i}\}_{i=1}^k) - d_{\mathcal{H}}(\widehat{\mathcal{D}}_T; \{\widehat{\mathcal{D}}_{S_i}\}_{i=1}^k)\right| \leq 2\sqrt{\frac{2}{m}\left(\log\frac{4k}{\delta} + d\log\frac{em}{d}\right)}$$

Note that multiple source domains do not increase the sample complexity too drastically: it is only the square root of a log term in Corollary. 3.1 where $k$ appears.

Similarly, we do not usually have access to the true error $\max_{i\in[k]}\varepsilon_{S_i}(h)$ on the source domains, but we can often have an estimate $(\max_{i\in[k]}\widehat{\varepsilon}_{S_i}(h))$ from training samples. We now provide a probabilistic guarantee to bound the difference between $\max_{i\in[k]}\varepsilon_{S_i}(h)$ and $\max_{i\in[k]}\widehat{\varepsilon}_{S_i}(h)$ uniformly for all $h \in \mathcal{H}$:

**Theorem 3.3.** Let $\{\mathcal{D}_{S_i}\}_{i=1}^k$ be $k$ source distributions over $\mathcal{X}$. Let $\mathcal{H}$ be a hypothesis class where $VC\dim(\mathcal{H}) = d$. If $\{\widehat{\mathcal{D}}_{S_i}\}_{i=1}^k$ are the empirical distributions of $\{\mathcal{D}_{S_i}\}_{i=1}^k$ generated with $m$ $i.i.d.$ samples from each domain, then, for $\epsilon > 0$, we have:

$$\Pr\left(\sup_{h\in\mathcal{H}}\left|\max_{i\in[k]}\varepsilon_{S_i}(h) - \max_{i\in[k]}\widehat{\varepsilon}_{S_i}(h)\right| \geq \epsilon\right) \leq 2k\left(\frac{me}{d}\right)^d \exp(-2m\epsilon^2)$$

Again, Thm. 3.3 can be proved by a combination of concentration inequalities and a reduction from infinite space to finite space, along with the subadditivity of the max function. Equivalently, we have the following corollary hold:

**Corollary 3.2.** Let $\{\mathcal{D}_{S_i}\}_{i=1}^k$ be $k$ source distributions over $\mathcal{X}$. Let $\mathcal{H}$ be a hypothesis class where $VC\dim(\mathcal{H}) = d$. If $\{\widehat{\mathcal{D}}_{S_i}\}_{i=1}^k$ are the empirical distributions of $\{\mathcal{D}_{S_i}\}_{i=1}^k$ generated with $m$ $i.i.d.$ samples from each domain, then, for $0 < \delta < 1$, with probability at least $1 - \delta$ (over the choice of samples), we have:

$$\sup_{h\in\mathcal{H}}\left|\max_{i\in[k]}\varepsilon_{S_i}(h) - \max_{i\in[k]}\widehat{\varepsilon}_{S_i}(h)\right| \leq \sqrt{\frac{1}{2m}\left(\log\frac{2k}{\delta} + d\log\frac{me}{d}\right)}$$

Combining Thm. 3.1 and Corollaries. 3.1, 3.2 and realizing that $VC\dim(\mathcal{H}\Delta\mathcal{H}) \leq 2VC\dim(\mathcal{H})$ (Anthony & Bartlett, 2009), we have the following theorem:

**Theorem 3.4.** Let $\mathcal{D}_T$ and $\{\mathcal{D}_{S_i}\}_{i=1}^k$ be the target distribution and $k$ source distributions over $\mathcal{X}$. Let $\mathcal{H}$ be a hypothesis class where $VC\dim(\mathcal{H}) = d$. If $\widehat{\mathcal{D}}_T$ and $\{\widehat{\mathcal{D}}_{S_i}\}_{i=1}^k$ are the empirical distributions of $\mathcal{D}_T$ and $\{\mathcal{D}_{S_i}\}_{i=1}^k$ generated with $m$ i.i.d. samples from each domain, then, for $0 < \delta < 1$, with probability at least $1 - \delta$ (over the choice of samples), we have:

$$\varepsilon_T(h) \leq \max_{i\in[k]} \widehat{\varepsilon}_{S_i}(h) + \sqrt{\frac{1}{2m}\left(\log\frac{4k}{\delta} + d\log\frac{me}{d}\right)} + \frac{1}{2}d_{\mathcal{H}\Delta\mathcal{H}}(\widehat{\mathcal{D}}_T; \{\widehat{\mathcal{D}}_{S_i}\}_{i=1}^k) + \sqrt{\frac{2}{m}\left(\log\frac{8k}{\delta} + 2d\log\frac{me}{2d}\right)} + \lambda$$

$$= \max_{i\in[k]} \widehat{\varepsilon}_{S_i}(h) + \frac{1}{2}d_{\mathcal{H}\Delta\mathcal{H}}(\widehat{\mathcal{D}}_T; \{\widehat{\mathcal{D}}_{S_i}\}_{i=1}^k) + O\left(\sqrt{\frac{1}{m}\left(\log\frac{k}{\delta} + d\log\frac{me}{d}\right)}\right) + \lambda \quad (2)$$

**Remark**. Thm. 3.4 has a nice interpretation for each term: the first term measures the worst case accuracy of hypothesis $h$ on the $k$ source domains, and the second term measures the discrepancy between the target domain and the $k$ source domains. For domain adaptation to succeed in the multiple sources setting, we have to expect these two terms to be small: we pick our hypothesis $h$ based on its source training errors, and it will generalize only if the discrepancy between sources and target is small. The third term bounds the additional error we may incur because of the possible bias from finite samples. The last term $\lambda$ is the optimal error we can hope to achieve. Hence, if $\lambda$ is large, one should not hope the generalization error to be small by training on the source domains. [1] It is also worth pointing out that these four terms appearing in the generalization bound also capture the tradeoff between using a rich hypothesis class $\mathcal{H}$ and a limited one as we discussed above: when using a richer hypothesis class, the first and the last terms in the bound will decrease, while the value of the second term will increase; on the other hand, choosing a limited hypothesis class can decrease the value of the second term, but we may incur additional source training errors and a large $\lambda$ due to the simplicity of $\mathcal{H}$. One interesting prediction implied by Thm. 3.4 is that the performance on the target domain depends on the worst empirical error among multiple source domains, i.e., it is not always beneficial to naively incorporate more source domains into training. As we will see in the experiment, this is indeed the case in many real-world problems. One alternative approach to obtain an upper bound for multiple source domains is to apply Thm. 2.1 repeatedly $k$ times, one for each source and target pair, followed by a union bound to combine them. It is easy to show that this approach can lead to a slightly tighter upper bound with the same asymptotic order in terms of $m$ and $k$ as the one in Thm. 3.4. However, as we will see in the next section, the bound in Thm. 3.4 provides a nice decoupling of the four terms so that minimizing the bound leads to two practical learning algorithms. As a comparison, the alternative bound cannot be minimized as it requires knowledge of unknown quantities $\lambda_i, \forall i \in [k]$, i.e., the optimal error on each pair of source and target domains.

**Comparison with Existing Bounds**    First, it is easy to see that, upto a multiplicative constant, our bound in (2) reduces to the one in Thm. 2.1 when there is only one source domain ($k = 1$). Hence Thm. 3.4 can be treated as a generalization of Thm. 2.1. Blitzer et al. (2008) give a generalization bound for semi-supervised multisource domain adaptation where, besides labeled instances from multiple source domains, the algorithm also has access to a fraction of labeled instances from the target domain. Although in general our bound and the one in (Blitzer et al., 2008, Thm. 3) are incomparable, it is instructive to see the connections and differences between them: on one hand, the multiplicative constants of the discrepancy measure and the optimal error in our bound are half of those in Blitzer et al. (2008)'s bound, leading to a tighter bound; on the other hand, because of the access to labeled instances from the target domain, their bound is expressed relative to the optimal error rate on the target domain, while ours is in terms of the empirical error on the source domain. Finally, thanks to our generalized definition of $d_{\mathcal{H}}(\mathcal{D}_T; \{\mathcal{D}_{S_i}\}_{i=1}^k)$, we do not need to manually specify the optimal combination vector $\alpha$ in (Blitzer et al., 2008, Thm. 3), which is unknown in practice. Mansour et al. (2009b) also give a generalization bound for multisource domain adaptation under the assumption that the target distribution is a mixture of the $k$ sources and the target hypothesis can be represented as a convex combination of the source hypotheses. While the distance measure we use assumes 0-1 loss function, their generalized discrepancy measure can also be applied for other loss functions (Mansour et al., 2009a;c;b).

---

[1]Of course it is still possible that $\varepsilon_T(h)$ is small while $\lambda$ is large, but in domain adaptation we do not have access to labeled samples from $\mathcal{D}_T$.

# 4 MULTISOURCE DOMAIN ADAPTATION WITH ADVERSARIAL NEURAL NETWORKS

In this section we shall describe a neural network based implementation to minimize the generalization bound we derive in Thm. 3.4. The key idea is to reformulate the generalization bound by a minimax saddle point problem and optimize it via adversarial training.

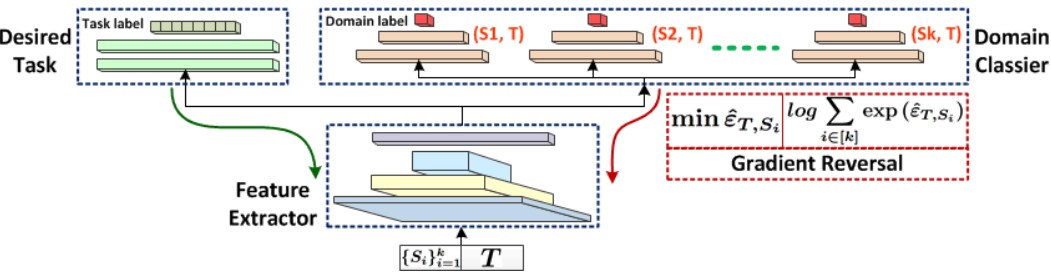

Figure 1: MDANs Network architecture. Feature extractor, domain classifier, and task learning are combined in one training process. Hard version: the source that achieves the minimum domain classification error is backpropagated with gradient reversal; Smooth version: all the domain classification risks over $k$ source domains are combined and backpropagated adaptively with gradient reversal.

Suppose we are given samples drawn from $k$ source domains $\{\mathcal{D}_{S_i}\}$, each of which contains $m$ instance-label pairs. Additionally, we also have access to unlabeled instances sampled from the target domain $\mathcal{D}_T$. Once we fix our hypothesis class $\mathcal{H}$, the last two terms in the generalization bound (2) will be fixed; hence we can only hope to minimize the bound by minimizing the first two terms, i.e., the maximum source training error and the discrepancy between source domains and target domain. The idea is to train a neural network to learn a representation with the following two properties: 1). indistinguishable between the $k$ source domains and the target domain; 2). informative enough for our desired task to succeed. Note that both requirements are necessary: without the second property, a neural network can learn trivial random noise representations for all the domains, and such representations cannot be distinguished by any discriminator; without the first property, the learned representation does not necessarily generalize to the unseen target domain. Taking these two properties into consideration, we propose the following optimization problem:

$$\text{minimize} \quad \max_{i \in [k]} \left( \widehat{\varepsilon}_{S_i}(h) + \frac{1}{2} d_{\mathcal{H} \Delta \mathcal{H}}(\widehat{\mathcal{D}}_T; \{\widehat{\mathcal{D}}_{S_i}\}_{i=1}^k) \right) \tag{3}$$

One key observation that leads to a practical approximation of $d_{\mathcal{H} \Delta \mathcal{H}}(\widehat{\mathcal{D}}_T; \{\widehat{\mathcal{D}}_{S_i}\}_{i=1}^k)$ from Ben-David et al. (2007) is that computing the discrepancy measure is closely related to learning a classifier that is able to disintuish samples from different domains:

$$d_{\mathcal{H} \Delta \mathcal{H}}(\widehat{\mathcal{D}}_T; \{\widehat{\mathcal{D}}_{S_i}\}_{i=1}^k) = \max_{i \in [k]} \left( 1 - 2 \min_{h \in \mathcal{H} \Delta \mathcal{H}} \left( \frac{1}{2m} \sum_{\mathbf{x} \sim \widehat{\mathcal{D}}_T} \mathbb{I}(h(\mathbf{x}) = 1) + \frac{1}{2m} \sum_{\mathbf{x} \sim \widehat{\mathcal{D}}_{S_i}} \mathbb{I}(h(\mathbf{x} = 0)) \right) \right)$$

Let $\widehat{\varepsilon}_{T,S_i}(h)$ be the empirical risk of hypothesis $h$ in the domain discriminating task. Ignoring the constant terms that do not affect the optimization formulation, moving the max operator out, we can reformulate (3) as:

$$\text{minimize} \quad \max_{i \in [k]} \left( \widehat{\varepsilon}_{S_i}(h) - \min_{h' \in \mathcal{H} \Delta \mathcal{H}} \widehat{\varepsilon}_{T,S_i}(h') \right) \tag{4}$$

The two terms in (4) exactly correspond to the two criteria we just proposed: the first term asks for an informative feature representation for our desired task to succeed, while the second term captures the notion of invariant feature representations between different domains.

Inspired by Ganin et al. (2016), we use the gradient reversal layer to effectively implement (4) by backpropagation. The network architecture is shown in Figure. 1. The pseudo-code is listed in Alg. 1 (the hard version). One notable drawback of the hard version in Alg. 1 is that in each iteration the

---

**Algorithm 1** Multiple Source Domain Adaptation via Adversarial Training

---

1: **for** $t = 1$ to $\infty$ **do**
2:     Sample $\{S_i^{(t)}\}_{i=1}^k$ and $T^{(t)}$ from $\{\widehat{\mathcal{D}}_{S_i}\}_{i=1}^k$ and $\widehat{\mathcal{D}}_T$, each of size $m$
3:     **for** $i = 1$ to $k$ **do**
4:         Compute $\widehat{\varepsilon}_i^{(t)} := \widehat{\varepsilon}_{S^{(t)}}(h) - \min_{h' \in \mathcal{H}\Delta\mathcal{H}} \widehat{\varepsilon}_{T^{(t)}, S^{(t)}}(h')$
5:         Compute $w_i^{(t)} := \exp(\widehat{\varepsilon}_i^{(t)})$
6:     **end for**
7:     **# Hard version**
8:     Select $i^{(t)} := \arg\max_{i \in [k]} \widehat{\varepsilon}_i^{(t)}$
9:     Update parameters via backpropagating gradient of $\widehat{\varepsilon}_{i^{(t)}}^{(t)}$
10:    **# Smoothed version**
11:    **for** $i = 1$ to $k$ **do**
12:        Normalize $w_i^{(t)} \leftarrow w_i^{(t)} / \sum_{i' \in [k]} w_{i'}^{(t)}$
13:    **end for**
14:    Update parameters via backpropagating gradient of $\sum_{i \in [k]} w_i^{(t)} \widehat{\varepsilon}_i^{(t)}$
15: **end for**

---

algorithm only updates its parameter based on the gradient from one of the $k$ domains. This is data inefficient and can waste our computational resources in the forward process. To improve this, we approximate the $\max$ function in (4) by the log-sum-exp function, which is a frequently used smooth approximation of the $\max$ function. Define $\widehat{\varepsilon}_i(h) := \widehat{\varepsilon}_{S_i}(h) - \min_{h' \in \mathcal{H}\Delta\mathcal{H}} \widehat{\varepsilon}_{T,S_i}(h')$:

$$\max_{i \in [k]} \widehat{\varepsilon}_i(h) \approx \frac{1}{\gamma} \log \sum_{i \in [k]} \exp(\gamma \widehat{\varepsilon}_i(h))$$

where $\gamma > 0$ is a parameter that controls the accuracy of this approximation. As $\gamma \to \infty$, $\frac{1}{\gamma} \log \sum_{i \in [k]} \exp(\gamma \widehat{\varepsilon}_i(h)) \to \max_{i \in [k]} \widehat{\varepsilon}_i(h)$. Correspondingly, we can formulate a smoothed version of (4) as:

$$\text{minimize} \quad \frac{1}{\gamma} \log \sum_{i \in [k]} \exp \left( \gamma(\widehat{\varepsilon}_{S_i}(h) - \min_{h' \in \mathcal{H}\Delta\mathcal{H}} \widehat{\varepsilon}_{T,S_i}(h')) \right) \tag{5}$$

During the optimization, (5) naturally provides an adaptive weighting scheme for the $k$ source domains depending on their relative error. Use $\theta$ to denote all the model parameters, then:

$$\frac{\partial}{\partial \theta} \frac{1}{\gamma} \log \sum_{i \in [k]} \exp \left( \gamma(\widehat{\varepsilon}_{S_i}(h) - \min_{h' \in \mathcal{H}\Delta\mathcal{H}} \widehat{\varepsilon}_{T,S_i}(h')) \right) = \sum_{i \in [k]} \frac{\exp \gamma \widehat{\varepsilon}_i(h)}{\sum_{i' \in [k]} \exp \gamma \widehat{\varepsilon}_{i'}(h)} \frac{\partial \widehat{\varepsilon}_i(h)}{\partial \theta} \tag{6}$$

The approximation trick not only smooths the objective, but also provides a principled and adaptive way to combine all the gradients from the $k$ source domains. In words, (6) says that the gradient of MDAN is a convex combination of the gradients from all the domains. The larger the error from one domain, the larger the combination weight in the ensemble. Formally, we show that (5) also corresponds to minimizing an upper bound of the generalization error on the target domain. But different from Thm. 3.4 where the worst case $\mathcal{H}$-divergence is considered, the following theorem characterizes the divergence between the target and multiple source domains using a weighted combination of divergences between the target domain and each source domain. As we will see in Sec. 5, the optimization problem (5) often leads to better generalizations in practice, which may partly be explained by the ensemble effect of multiple sources implied by the upper bound.

**Theorem 4.1.** Let $\mathcal{D}_T$ and $\{\mathcal{D}_{S_i}\}_{i=1}^k$ be the target distribution and $k$ source distributions over $\mathcal{X}$. Let $\mathcal{H}$ be a hypothesis class where $VC\dim(\mathcal{H}) = d$. If $\widehat{\mathcal{D}}_T$ and $\{\widehat{\mathcal{D}}_{S_i}\}_{i=1}^k$ are the empirical distributions of $\mathcal{D}_T$ and $\{\mathcal{D}_{S_i}\}_{i=1}^k$ generated with $m$ i.i.d. samples from each domain, then, $\forall \alpha \in \mathbb{R}_+^k, \sum_{i \in [k]} \alpha_i = 1$, for $0 < \delta < 1$, with probability at least $1 - \delta$ (over the choice of samples), we have:

$$\varepsilon_T(h) \leq \sum_{i \in [k]} \alpha_i \cdot \left( \widehat{\varepsilon}_{S_i}(h) + \frac{1}{2} d_{\mathcal{H}\Delta\mathcal{H}}(\widehat{\mathcal{D}}_T; \widehat{\mathcal{D}}_{S_i}) \right) + O\left( \sqrt{\frac{1}{m} \left( \log \frac{k}{\delta} + d \log \frac{me}{d} \right)} \right) + \lambda_\alpha \tag{7}$$

where $\lambda_\alpha$ is a constant that only depends on $\mathcal{H}$.

By a proper choice of $\alpha$ in the upper bound of Thm. 4.1, we obtain the following upper bound that is minimized by the optimization problem (5).

**Theorem 4.2.** Choose $\alpha_i = \exp(\widehat{\varepsilon}_i(h))/\sum_{j\in[k]}\exp(\widehat{\varepsilon}_j(h))$ with $\widehat{\varepsilon}_i(h) := \widehat{\varepsilon}_{S_i}(h) + \frac{1}{2}d_{\mathcal{H}\Delta\mathcal{H}}(\widehat{\mathcal{D}}_T; \widehat{\mathcal{D}}_{S_i})$, we have

$$\varepsilon_T(h) \leq \log \sum_{i\in[k]} \exp(\widehat{\varepsilon}_i(h)) + O\left(\sqrt{\frac{1}{m}\left(\log\frac{k}{\delta} + d\log\frac{me}{d}\right)}\right) + \lambda^* \tag{8}$$

where $\lambda^*$ is a constant that only depends on $\mathcal{H}$.

**Remark.** It is not hard to see that, up to constant that does not depend on the training errors on multiple domains, the upper bound given by Thm. 4.2 is tighter than that of Thm. 3.4. We also note that both sample complexity bounds given in Thm. 3.4 and Thm. 4.2 are optimal in terms of the number of training instances $m$ in each source domain, as it matches the $\Omega(\sqrt{1/m})$ lower bound in the non-realizable binary classification scenario (Mohri et al., 2012, Thm. 3.7).

We summarize this algorithm in the smoothed version of Alg. 1. Note that both algorithms, including the hard version and the smoothed version, reduce to the DANN algorithm (Ganin et al., 2016) when there is only one source domain.

## 5 EXPERIMENTS

We evaluate both hard and soft MDANs and compare them with state-of-the-art methods on three real-world datasets: the Amazon benchmark dataset (Chen et al., 2012) for sentiment analysis, a digit classification task that includes 4 datasets: MNIST (LeCun et al., 1998), MNIST-M (Ganin et al., 2016), SVHN (Netzer et al., 2011), and SynthDigits (Ganin et al., 2016), and a public, large-scale image dataset on vehicle counting from multiple city cameras (Zhang et al., 2017). Details about network architecture and training parameters of proposed and baseline methods, and detailed dataset description will be introduced in the appendix.

### 5.1 AMAZON REVIEWS

Domains within the dataset consist of reviews on a specific kind of product (Books, DVDs, Electronics, and Kitchen appliances). Reviews are encoded as 5000 dimensional feature vectors of unigrams and bigrams, with binary labels indicating sentiment. We conduct 4 experiments: for each of them, we pick one product as target domain and the rest as source domains. Each source domain has 2000 labeled examples, and the target test set has 3000 to 6000 examples. During training, we randomly sample the same number of unlabeled target examples as the source examples in each mini-batch. We implement the Hard-Max and Soft-Max methods according to Alg. 1, and compare them with three baselines: MLPNet, marginalized stacked denoising autoencoders (mSDA) (Chen et al., 2012), and DANN (Ganin et al., 2016). DANN cannot be directly applied in multiple source domains setting. In order to make a comparison, we use two protocols. The first one is to combine all the source domains into a single one and train it using DANN, which we denote as Combine-DANN. The second protocol is to train multiple DANNs separately, where each one corresponds to a source-target pair. Among all the DANNs, we report the one achieving the best performance on the target domain. We denote this experiment as Best-Single-DANN. For fair comparison, all these models are built on the same basic network structure with one input layer (5000 units) and three hidden layers (1000, 500, 100 units).

Table 1: Sentiment classification accuracy.

| Train/Test | MLPNet | mSDA | Best-Single-DANN | Combine-DANN | MDANs | |
| --- | --- | --- | --- | --- | --- | --- |
| | | | | | Hard-Max | Soft-Max |
| **D+E+K/B** | 0.7655 | 0.7698 | 0.7650 | 0.7789 | 0.7845 | **0.7863** |
| **B+E+K/D** | 0.7588 | 0.7861 | 0.7732 | 0.7886 | 0.7797 | **0.8065** |
| **B+D+K/E** | 0.8460 | 0.8198 | 0.8381 | 0.8491 | 0.8483 | **0.8534** |
| **B+D+E/K** | 0.8545 | 0.8426 | 0.8433 | **0.8639** | 0.8580 | 0.8626 |

Table 2: $p$-values under Wilcoxon test.

| | MLPNet | mSDA | Best-Single-DANN | Combine-DANN | Hard-Max |
|---|---|---|---|---|---|
| | Soft-Max | Soft-Max | Soft-Max | Soft-Max | Soft-Max |
| **B** | 0.550 | 0.101 | 0.521 | 0.013 | 0.946 |
| **D** | 0.000 | 0.072 | 0.000 | 0.051 | 0.000 |
| **E** | 0.066 | 0.000 | 0.097 | 0.150 | 0.022 |
| **K** | 0.306 | 0.001 | 0.001 | 0.239 | 0.008 |

**Results and Analysis**   We show the accuracy of different methods in Table 1. Clearly, Soft-Max significantly outperforms all other methods in most settings. When Kitchen is the target domain, cDANN performs slightly better than Soft-Max, and all the methods perform close to each other. Hard-Max is typically slightly worse than Soft-Max. This is mainly due to the low data-efficiency of the Hard-Max model (Section 4, Eq. 4, Eq. 5). We argue that with more training iterations, the performance of Hard-Max can be further improved. These results verify the effectiveness of MDANs for multisource domain adaptation. To validate the statistical significance of the results, we run a non-parametric Wilcoxon signed-ranked test for each task to compare Soft-Max with the other competitors, as shown in Table 2. Each cell corresponds to the $p$-value of a Wilcoxon test between Soft-Max and one of the other methods, under the null hypothesis that the two paired samples have the same mean. From these p-values, we see Soft-Max is convincingly better than other methods.

## 5.2   DIGITS DATASETS

Following the setting in (Ganin et al., 2016), we combine four popular digits datasets (MNIST, MNIST-M, SVHN, and SynthDigits) to build the multisource domain dataset. We take each of MNIST-M, SVHN, and MNIST as target domain in turn, and the rest as sources. Each source domain has $20,000$ labeled images and the target test set has $9,000$ examples.

**Baselines**   We compare Hard-Max and Soft-Max of MDANs with ten baselines: i). *Best-Single-Source*. A basic network trained on each source domain ($20,000$ images) without domain adaptation and tested on the target domain. Among the three models, we report the one achieves the best performance on the test set. ii). *Combine-Source*. A basic network trained on a combination of three source domains ($20,000$ images for each) without domain adaptation and tested on the target domain. iii). *Best-Single-DANN*. We train DANNs (Ganin et al., 2016) on each source-target domain pair ($20,000$ images for each source) and test it on target. Again, we report the best score among the three. iv). *Combine-DANN*. We train a single DANN on a combination of three source domains ($20,000$ images for each). v). *Best-Single-ADDA*. We train ADDA (Tzeng et al., 2017) on each source-target domain pair ($20,000$ images for each source) and test it on the target domain. We report the best accuracy among the three. ADDA is an unsupervised adversarial adaptation method, which first learns a discriminative representation using the labels in the source domain and then a separate encoding that maps the target data to the same space using an asymmetric mapping learned through a domain-adversarial loss. vi).*Combine-ADDA*. We train ADDA on a combination of three source domains ($20,000$ images for each). vii). *Best-Single-MTAE*. We train MTAE (Ghifary et al., 2015) on each source-target domain pair ($20,000$ images for each source) and test it on the target domain. We report the best accuracy among the three. MTAE is feature learning algorithm that extends the standard denoising autoencoder framework by substituting corruption with naturally occurring inter-domain variability in the appearance of objects. It learns to transform the original image into analogs in multiple related domains, thereby learns features that are robust to variations across domains. The learned features are then used as inputs to the classifier. viii). *Combine-MTAE*. We train MTAE on a combination of three source domains ($20,000$ images for each). ix). *MDAC*. MDAC (Zhang et al., 2015) is a multiple source domain adaptation algorithm that explores causal models to represent the relationship between the features $X$ and class label $Y$. It models $P_X|P_Y$ (the process to generate effect $X$ from cause $Y$) on the target domain as a linear mixture of those on source domains, and estimate all involved parameters by matching the target-domain feature distribution. As MDAC is designed for multiple source domain adaptation, we directly train MDAC on a combination of three source domains. x). *Target-only*. It is the basic network trained and tested

on the target data. It serves as an upper bound of DA algorithms. All the MDANs and baseline methods are built on the same basic network structure to put them on a equal footing.

**Results and Analysis**  The classification accuracy is shown in Table 3. The results show that MDAN outperforms all the baselines in the first two experiments and is comparable with Best-Single-DANN in the third experiment. For the combined sources, MDANs always perform better than the source-only baseline (MDANs vs. Combine-Source). However, a naive combination of different training datasets can sometimes even decrease the performance of the baseline methods. This conclusion comes from three observations: First, directly training DANN on a combination of multiple sources leads to worse results than the source-only baseline (Combine-DANN vs. Combine-Source); Second, The performance of Combine-DANN can be even worse than the Best-Single-DANN (the first and third experiments); Third, directly training DANN on a combination of multiple sources always has lower accuracy compared with our approach (Combine-DANN vs. MDANs). We have similar observations for ADDA and MTAE. Such observations verify that the domain adaptation methods designed for single source lead to suboptimal solutions when applied to multiple sources. It also verifies the necessity and superiority of MDAN for multiple source adaptation. Though MDAC is designed for multiple source domain adaptation, it has obviously lower accuracy than MDANs. Furthermore, we observe that adaptation to the SVHN dataset (the third experiment) is hard. In this case, increasing the number of source domains does not help. We conjecture this is due to the large dissimilarity between the SVHN data to the others. Surprisingly, using a single domain (best-Single DANN) in this case achieves the best result. This indicates that in domain adaptation the quality of data (how close to the target data) is much more important than the quantity (how many source domains). As a conclusion, this experiment further demonstrates the effectiveness of MDANs when there are multiple source domains available, where a naive combination of multiple sources using DANN may hurt generalization.

Table 3: Accuracy on digit classification. Mt: MNIST; Mm: MNIST-M, Sv: SVHN, Sy: SynthDigits.

| Method | Sv+Mm+Sy/Mt | Mt+Sv+Sy/Mm | Mm+Mt+Sy/Sv |
|---|---|---|---|
| **Best-Single-Source** | 0.964 | 0.519 | 0.814 |
| **Best-Single-DANN** | 0.967 | 0.591 | **0.818** |
| **Best-Single-ADDA** | 0.968 | 0.657 | 0.800 |
| **Best-Single-MTAE** | 0.862 | 0.534 | 0.703 |
| **Combine-Source** | 0.938 | 0.561 | 0.771 |
| **MDAC** | 0.755 | 0.563 | 0.604 |
| **Combine-DANN** | 0.925 | 0.651 | 0.776 |
| **Combine-ADDA** | 0.927 | 0.682 | 0.804 |
| **Combine-MTAE** | 0.821 | 0.596 | 0.701 |
| **MDAN-Hard-Max** | 0.976 | 0.663 | 0.802 |
| **MDAN-Soft-Max** | **0.979** | **0.687** | 0.816 |
| **Target-only** | 0.987 | 0.901 | 0.898 |

Table 4: Counting error statistics. S is the number of source cameras; T is the target camera id.

| S | T | MDANs | | DANN | FCN | T | MDANs | | DANN | FCN |
|---|---|---|---|---|---|---|---|---|---|---|
| | | Hard-Max | Soft-Max | | | | Hard-Max | Soft-Max | | |
| 2 | A | 1.8101 | **1.7140** | 1.9490 | 1.9094 | B | 2.5059 | **2.3438** | 2.5218 | 2.6528 |
| 3 | A | 1.3276 | **1.2363** | 1.3683 | 1.5545 | B | 1.9092 | **1.8680** | 2.0122 | 2.4319 |
| 4 | A | 1.3868 | **1.1965** | 1.5520 | 1.5499 | B | **1.7375** | 1.8487 | 2.1856 | 2.2351 |
| 5 | A | 1.4021 | **1.1942** | 1.4156 | 1.7925 | B | 1.7758 | **1.6016** | 1.7228 | 2.0504 |
| 6 | A | 1.4359 | **1.2877** | 2.0298 | 1.7505 | B | 1.5912 | **1.4644** | 1.5484 | 2.2832 |
| 7 | A | 1.4381 | **1.2984** | 1.5426 | 1.7646 | B | 1.5989 | **1.5126** | 1.5397 | 1.7324 |

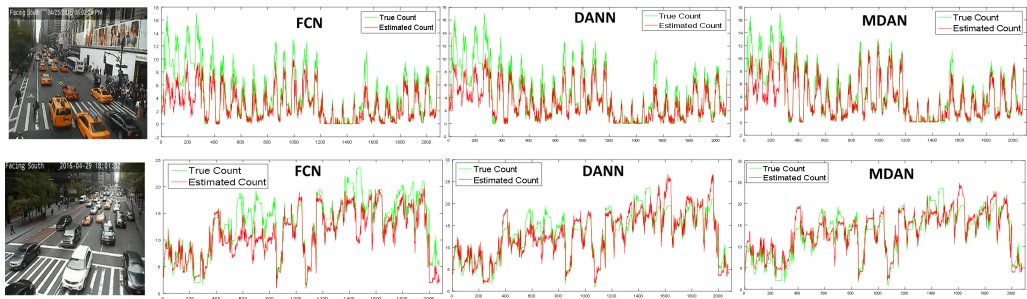

Figure 2: Counting results for target camera A (first row) and B (second row). X-frames; Y-Counts.

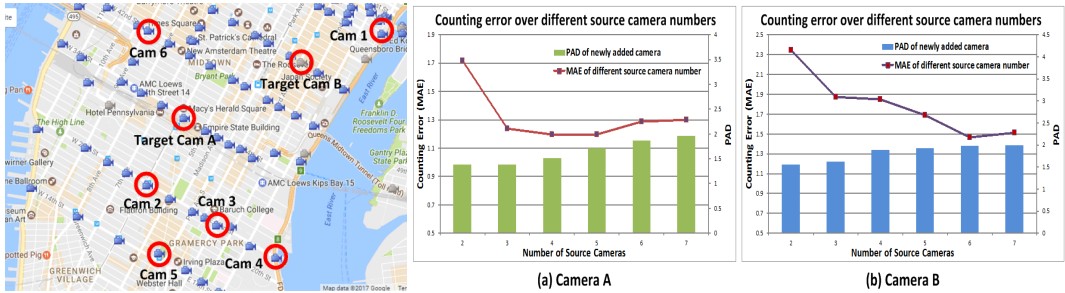

Figure 3: Source&target camera map.     Figure 4: Counting error over different source numbers.

## 5.3 WEBCAMT VEHICLE COUNTING DATASET

WebCamT is a public dataset for vehicle counting from large-scale city camera videos, which has low resolution ($352 \times 240$), low frame rate (1 frame/second), and high occlusion. It has $60,000$ frames annotated with vehicle bounding box and count, divided into training and testing sets, with $42,200$ and $17,800$ frames, respectively. Here we demonstrate the effectiveness of MDANs to count vehicles from an unlabeled target camera by adapting from multiple labeled source cameras: we select $8$ cameras that each has more than $2,000$ labeled images for our evaluations. As shown in Fig. 3, they are located in different intersections of the city with different scenes. Among these $8$ cameras, we randomly pick two cameras and take each camera as the target camera, with the other $7$ cameras as sources. We compute the proxy $\mathcal{A}$-distance (PAD) (Ben-David et al., 2007) between each source camera and the target camera to approximate the divergence between them. We then rank the source cameras by the PAD from low to high and choose the first $k$ cameras to form the $k$ source domains. Thus the proposed methods and baselines can be evaluated on different numbers of sources (from 2 to 7). We implement the Hard-Max and Soft-Max MDANs according to Alg. 1, based on the basic vehicle counting network FCN (Zhang et al., 2017). We compare our method with two baselines: FCN (Zhang et al., 2017), a basic network without domain adaptation, and DANN (Ganin et al., 2016), implemented on top of the same basic network. We record mean absolute error (MAE) between true count and estimated count.

**Results and Analysis** The counting error of different methods is compared in Table 4. The Hard-Max version achieves lower error than DANN and FCN in most settings for both target cameras. The Soft-Max approximation outperforms all the baselines and the Hard-Max in most settings, demonstrating the effectiveness of the smooth and adaptative approximation. The lowest MAE achieved by Soft-Max is $1.1942$. Such MAE means that there is only around one vehicle miscount for each frame (the average number of vehicles in one frame is around 20). Fig. 2 shows the counting results of Soft-Max for the two target cameras under the 5 source cameras setting. We can see that the proposed method accurately counts the vehicles of each target camera for long time sequences. Does adding more source cameras always help improve the performance on the target camera? To answer this question, we analyze the counting error when we vary the number of source cameras as shown in Fig. 4. From the curves, we see the counting error goes down with more source cameras at the beginning, while it goes up when more sources are added at the end. This phenomenon corresponds to the prediction implied by Thm. 3.4 (the last remark in Section 3): the performance on the target

domain depends on the worst empirical error among multiple source domains, i.e., it is not always beneficial to naively incorporate more source domains into training. To illustrate this prediction better, we show the PAD of the newly added camera (when the source number increases by one) in Fig. 4. By observing the PAD and the counting error, we see the performance on the target can degrade when the newly added source camera has large divergence from the target camera.

## 6 CONCLUSION

We derive a new generalization bound for DA under the setting of multiple source domains with labeled instances and one target domain with unlabeled instances. The new bound has interesting interpretation and reduces to an existing bound when there is only one source domain. Following our theoretical results, we propose MDANs to learn feature representations that are invariant under multiple domain shifts while at the same time being discriminative for the learning task. Both hard and soft versions of MDANs are generalizations of the popular DANN to the case when multiple source domains are available. Empirically, MDANs outperform the state-of-the-art DA methods on three real-world datasets, including a sentiment analysis task, a digit classification task, and a visual vehicle counting task, demonstrating its effectiveness for multisource domain adaptation.

## ACKNOWLEDGEMENTS

The authors would like to thank the anonymous reviewers for thoughtful discussions and comments.

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

# A  OUTLINE

Organization of the appendix: 1). For the convenience of exposition in showing our technical proofs, we first introduce the technical tools that will be used during our proofs in Sec. B. 2). We provide detailed proofs for all the claims, lemmas and theorems presented in the main paper in Sec. C. 3). We describe more experimental details in Sec. D, including dataset description, network architecture and training parameters of the proposed and baseline methods, and more analysis of the experimental results. 4). We introduce and discuss more related work about domain adaptation in Sec. E.

# B  TECHNICAL TOOLS

**Definition B.1** (Growth function). The *growth function* $\Pi_{\mathcal{H}} : \mathbb{N} \rightarrow \mathbb{N}$ for a hypothesis class $\mathcal{H}$ is defined by:
$$\forall m \in \mathbb{N}, \quad \Pi_{\mathcal{H}}(m) = \max_{X_m \subseteq \mathcal{X}} |\{(h(x_1), \dots, h(x_m)) \mid h \in \mathcal{H}\}|$$
where $X_m = \{x_1, \dots, x_m\}$ is a subset of $\mathcal{X}$ with size $m$.

Roughly, the growth function $\Pi_{\mathcal{H}}(m)$ computes the maximum number of distinct ways in which $m$ points can be classified using hypothesis in $\mathcal{H}$. A closely related concept is the *Vapnik–Chervonenkis dimension* (VC dimension) (Vapnik, 1998):

**Definition B.2** (VC dimension). The VC-dimension of a hypothesis class $\mathcal{H}$ is defined as:
$$VC\mathrm{dim}(\mathcal{H}) = \max\{m : \Pi_{\mathcal{H}}(m) = 2^m\}$$

A well-known result relating $VC\mathrm{dim}(\mathcal{H})$ and the growth function $\Pi_{\mathcal{H}}(m)$ is the Sauer's lemma:

**Lemma B.1** (Sauer's lemma). Let $\mathcal{H}$ be a hypothesis class with $VC\mathrm{dim}(\mathcal{H}) = d$. Then, for $m \geq d$, the following inequality holds:
$$\Pi_{\mathcal{H}}(m) \leq \sum_{i=0}^{d} \binom{m}{i} \leq \left(\frac{em}{d}\right)^d$$

The following concentration inequality will be used:

**Theorem B.1** (Hoeffding's inequality). Let $X_1, \dots, X_n$ be independent random variables where each $X_i$ is bounded by the interval $[a_i, b_i]$. Define the empirical mean of these random variables by $\bar{X} := \frac{1}{n} \sum_{i=1}^{n} X_i$, then $\forall \varepsilon > 0$:
$$\Pr\left(\left|\bar{X} - \mathbb{E}[\bar{X}]\right| \geq \varepsilon\right) \leq 2 \exp\left(-\frac{2n^2 \varepsilon^2}{\sum_{i=1}^{n}(b_i - a_i)^2}\right)$$

The VC inequality allows us to give a uniform bound on the binary classification error of a hypothesis class $\mathcal{H}$ using growth function:

**Theorem B.2** (VC inequality). Let $\Pi_{\mathcal{H}}$ be the growth function of hypothesis class $\mathcal{H}$. For $h \in \mathcal{H}$, let $\varepsilon(h)$ be the true risk of $h$ w.r.t. the generation distribution $\mathcal{D}$ and the true labeling function $h^*$. Similarly, let $\hat{\varepsilon}_n(h)$ be the empirical risk on a random $i.i.d.$ sample containing $n$ instances from $\mathcal{D}$, then, for $\forall \varepsilon > 0$, the following inequality hold:
$$\Pr\left(\sup_{h \in \mathcal{H}} |\varepsilon(h) - \hat{\varepsilon}_n(h)| \geq \varepsilon\right) \leq 8\Pi_{\mathcal{H}}(n) \exp\left(-n\varepsilon^2/32\right)$$

Although the above theorem is stated for binary classification error, we can extend it to any bounded error. This will only change the multiplicative constant of the bound.

# C  PROOFS

For all the proofs presented here, the following lemma shown by Blitzer et al. (2008) will be repeatedly used:

**Lemma C.1** ((Blitzer et al., 2008)). $\forall h, h' \in \mathcal{H}, \quad |\varepsilon_S(h, h') - \varepsilon_T(h, h')| \leq \frac{1}{2} d_{\mathcal{H}\Delta\mathcal{H}}(\mathcal{D}_S, \mathcal{D}_T)$.

## C.1  PROOF OF THM. 3.1

One technical lemma we will frequently use to prove Thm. 3.1 is the triangular inequality w.r.t. $\varepsilon_{\mathcal{D}}(h), \forall h \in \mathcal{H}$:

**Lemma C.2.** For any hypothesis class $\mathcal{H}$ and any distribution $\mathcal{D}$ on $\mathcal{X}$, the following triangular inequality holds:

$$\forall h, h', f \in \mathcal{H}, \quad \varepsilon_{\mathcal{D}}(h, h') \leq \varepsilon_{\mathcal{D}}(h, f) + \varepsilon_{\mathcal{D}}(f, h')$$

*Proof.*

$$\varepsilon_{\mathcal{D}}(h, h') = \mathbb{E}_{\mathbf{x} \sim \mathcal{D}}[|h(\mathbf{x}) - h'(\mathbf{x})|] \leq \mathbb{E}_{\mathbf{x} \sim \mathcal{D}}[|h(\mathbf{x}) - f(\mathbf{x})| + |f(\mathbf{x}) - f(\mathbf{x})|] = \varepsilon_{\mathcal{D}}(h, f) + \varepsilon_{\mathcal{D}}(f, h')$$

∎

Now we are ready to prove Thm. 3.1:

**Theorem 3.1.** $\varepsilon_T(h) \leq \max_{i \in [k]} \varepsilon_{S_i}(h) + \frac{1}{2} d_{\mathcal{H} \Delta \mathcal{H}}(\mathcal{D}_T; \{\mathcal{D}_{S_i}\}_{i=1}^k) + \lambda.$

*Proof.* $\forall h \in \mathcal{H}$, define $i_h := \arg \max_{i \in [k]} \varepsilon_{S_i}(h, h^*)$:

$$\begin{aligned}
\varepsilon_T(h) &\leq \varepsilon_T(h^*) + \varepsilon_T(h, h^*) \\
&= \varepsilon_T(h^*) + \varepsilon_T(h, h^*) - \max_{i \in [k]} \varepsilon_{S_i}(h, h^*) + \max_{i \in [k]} \varepsilon_{S_i}(h, h^*) \\
&\leq \varepsilon_T(h^*) + |\varepsilon_T(h, h^*) - \varepsilon_{S_{i_h}}(h, h^*)| + \varepsilon_{S_{i_h}}(h, h^*) \\
&\leq \varepsilon_T(h^*) + \frac{1}{2} d_{\mathcal{H} \Delta \mathcal{H}}(\mathcal{D}_T, \mathcal{D}_{S_{i_h}}) + \varepsilon_{S_{i_h}}(h, h^*) \\
&\leq \varepsilon_T(h^*) + \frac{1}{2} d_{\mathcal{H} \Delta \mathcal{H}}(\mathcal{D}_T; \{\mathcal{D}_{S_i}\}_{i=1}^k) + \varepsilon_{S_{i_h}}(h, h^*) \\
&\leq \varepsilon_T(h^*) + \frac{1}{2} d_{\mathcal{H} \Delta \mathcal{H}}(\mathcal{D}_T; \{\mathcal{D}_{S_i}\}_{i=1}^k) + \varepsilon_{S_{i_h}}(h) + \varepsilon_{S_{i_h}}(h^*) \\
&\leq \varepsilon_T(h^*) + \frac{1}{2} d_{\mathcal{H} \Delta \mathcal{H}}(\mathcal{D}_T; \{\mathcal{D}_{S_i}\}_{i=1}^k) + \max_{i \in [k]} \varepsilon_{S_i}(h) + \max_{i \in [k]} \varepsilon_{S_i}(h^*) \\
&= \max_{i \in [k]} \varepsilon_{S_i}(h) + \lambda + \frac{1}{2} d_{\mathcal{H} \Delta \mathcal{H}}(\mathcal{D}_T; \{\mathcal{D}_{S_i}\}_{i=1}^k)
\end{aligned}$$

The first and the fifth inequalities are due to the triangle inequality, and the third inequality is based on Lemma C.1. The second holds due to the property of $|\cdot|$ and the others follow by the definition of $\mathcal{H}$-divergence. ∎

## C.2  PROOF OF THM. 3.2

**Theorem 3.2.** Let $\mathcal{D}_T$ and $\{\mathcal{D}_{S_i}\}_{i=1}^k$ be the target distribution and $k$ source distributions over $\mathcal{X}$. Let $\mathcal{H}$ be a hypothesis class where $VC\mathrm{dim}(\mathcal{H}) = d$. If $\widehat{\mathcal{D}}_T$ and $\{\widehat{\mathcal{D}}_{S_i}\}_{i=1}^k$ are the empirical distributions of $\mathcal{D}_T$ and $\{\mathcal{D}_{S_i}\}_{i=1}^k$ generated with $m$ *i.i.d.* samples from each domain, then for $\epsilon > 0$, we have:

$$\Pr\left(\left| d_{\mathcal{H}}(\mathcal{D}_T; \{\mathcal{D}_{S_i}\}_{i=1}^k) - d_{\mathcal{H}}(\widehat{\mathcal{D}}_T; \{\widehat{\mathcal{D}}_{S_i}\}_{i=1}^k) \right| \geq \epsilon\right) \leq 4k \left(\frac{em}{d}\right)^d \exp\left(-m\epsilon^2/8\right)$$

*Proof.*

$$\Pr\left(\left|d_{\mathcal{H}}(\mathcal{D}_T; \{\mathcal{D}_{S_i}\}_{i=1}^k) - d_{\mathcal{H}}(\hat{\mathcal{D}}_T; \{\hat{\mathcal{D}}_{S_i}\}_{i=1}^k)\right| \geq \epsilon\right)$$

$$= \Pr\left(\left|\max_{i\in[k]}\sup_{A\in\mathcal{A}_{\mathcal{H}}}|\Pr_{\mathcal{D}_T}(A) - \Pr_{\mathcal{D}_{S_i}}(A)| - \max_{i\in[k]}\sup_{A\in\mathcal{A}_{\mathcal{H}}}|\Pr_{\hat{\mathcal{D}}_T}(A) - \Pr_{\hat{\mathcal{D}}_{S_i}}(A)|\right| \geq \frac{\epsilon}{2}\right)$$

$$\leq \Pr\left(\max_{i\in[k]}\sup_{A\in\mathcal{A}_{\mathcal{H}}}\left||\Pr_{\mathcal{D}_T}(A) - \Pr_{\mathcal{D}_{S_i}}(A)| - |\Pr_{\hat{\mathcal{D}}_T}(A) - \Pr_{\hat{\mathcal{D}}_{S_i}}(A)|\right| \geq \frac{\epsilon}{2}\right)$$

$$= \Pr\left(\exists i\in[k], \exists A\in\mathcal{A}_{\mathcal{H}} : \left||\Pr_{\mathcal{D}_T}(A) - \Pr_{\mathcal{D}_{S_i}}(A)| - |\Pr_{\hat{\mathcal{D}}_T}(A) - \Pr_{\hat{\mathcal{D}}_{S_i}}(A)|\right| \geq \frac{\epsilon}{2}\right)$$

$$\leq \sum_{i=1}^k \Pr\left(\exists A\in\mathcal{A}_{\mathcal{H}} : \left||\Pr_{\mathcal{D}_T}(A) - \Pr_{\mathcal{D}_{S_i}}(A)| - |\Pr_{\hat{\mathcal{D}}_T}(A) - \Pr_{\hat{\mathcal{D}}_{S_i}}(A)|\right| \geq \frac{\epsilon}{2}\right)$$

$$\leq \sum_{i=1}^k \Pr\left(\exists A\in\mathcal{A}_{\mathcal{H}} : |\Pr_{\mathcal{D}_T}(A) - \Pr_{\hat{\mathcal{D}}_T}(A)| + |\Pr_{\mathcal{D}_{S_i}}(A) - \Pr_{\hat{\mathcal{D}}_{S_i}}(A)| \geq \frac{\epsilon}{2}\right)$$

$$\leq 2k\Pr\left(\exists A\in\mathcal{A}_{\mathcal{H}} : |\Pr_{\mathcal{D}_T}(A) - \Pr_{\hat{\mathcal{D}}_T}(A)| \geq \frac{\epsilon}{4}\right)$$

$$\leq 2k\cdot\Pi_{\mathcal{A}_{\mathcal{H}}}(m)\Pr\left(|\Pr_{\mathcal{D}_T}(A) - \Pr_{\hat{\mathcal{D}}_T}(A)| \geq \frac{\epsilon}{4}\right)$$

$$\leq 2k\cdot\Pi_{\mathcal{A}_{\mathcal{H}}}(m)\cdot 2\exp(-2m\epsilon^2/16)$$

$$\leq 4k\left(\frac{em}{d}\right)^d \exp(-m\epsilon^2/8)$$

The first inequality holds due to the sub-additivity of the max function, and the second inequality is due to the union bound. The third inequality holds because of the triangle inequality, and we use the averaging argument to establish the fourth inequality. The fifth inequality is an application of the VC-inequality, and the sixth is by the Hoeffding's inequality. Finally, we use the Sauer's lemma to prove the last inequality. ∎

### C.3 Proof of Thm. 3.3

We now show the detailed proof of Thm. 3.3.

*Proof.*

$$\Pr\left(\sup_{h\in\mathcal{H}}\left|\max_{i\in[k]}\varepsilon_{S_i}(h) - \max_{i\in[k]}\hat{\varepsilon}_{S_i}(h)\right| \geq \epsilon\right) \leq \Pr\left(\sup_{h\in\mathcal{H}}\max_{i\in[k]}|\varepsilon_{S_i}(h) - \hat{\varepsilon}_{S_i}(h)| \geq \epsilon\right)$$

$$= \Pr\left(\max_{i\in[k]}\sup_{h\in\mathcal{H}}|\varepsilon_{S_i}(h) - \hat{\varepsilon}_{S_i}(h)| \geq \epsilon\right)$$

$$\leq \sum_{i=1}^k \Pr\left(\sup_{h\in\mathcal{H}}|\varepsilon_{S_i}(h) - \hat{\varepsilon}_{S_i}(h)| \geq \epsilon\right)$$

$$\leq k\cdot\Pi_{\mathcal{H}}(m)\Pr\left(|\varepsilon_{S_i}(h) - \hat{\varepsilon}_{S_i}(h)| \geq \epsilon\right)$$

$$\leq k\cdot\Pi_{\mathcal{H}}(m)\cdot 2\exp(-2m\epsilon^2)$$

$$\leq 2k\left(\frac{me}{d}\right)^d \exp(-2m\epsilon^2)$$

Again, the first inequality is due to the subadditivity of the max function, and the second inequality holds due to the union bound. We apply the VC-inequality to bound the third inequality, and Hoeffding's inequality to bound the fourth. Again, the last one is due to Sauer's lemma. ∎

## C.4 Derivation of the Discrepancy Distance as Classification Error

We show that the $\mathcal{H}$-divergence is equivalent to a binary classification accuracy in discriminating instances from different domains. Suppose $\mathcal{A}_{\mathcal{H}}$ is symmetric, i.e., $A \in \mathcal{A}_{\mathcal{H}} \Leftrightarrow \mathcal{X} \backslash A \in \mathcal{A}_{\mathcal{H}}$, and we have samples $\{S_i\}_{i=1}^k$ and $T$ from $\{\mathcal{D}_{S_i}\}_{i=1}^k$ and $\mathcal{D}_T$ respectively, each of which is of size $m$, then:

$$
\begin{aligned}
d_{\mathcal{H}\Delta\mathcal{H}}(\hat{\mathcal{D}}_T; \{\hat{\mathcal{D}}_{S_i}\}_{i=1}^k) &= \max_{i\in[k]} \sup_{A \in \mathcal{A}_{\mathcal{H}\Delta\mathcal{H}}} |\Pr_{\hat{\mathcal{D}}_T}(A) - \Pr_{\hat{\mathcal{D}}_{S_i}}(A)| \\
&= \max_{i\in[k]} \sup_{h\in\mathcal{H}\Delta\mathcal{H}} |\Pr_{\mathbf{x}\sim\hat{\mathcal{D}}_T}(h(\mathbf{x})=1) - \Pr_{\mathbf{x}\sim\hat{\mathcal{D}}_{S_i}}(h(\mathbf{x}=1))| \\
&= \max_{i\in[k]} \sup_{h\in\mathcal{H}\Delta\mathcal{H}} 1 - \left( \Pr_{\mathbf{x}\sim\hat{\mathcal{D}}_T}(h(\mathbf{x})=1) + \Pr_{\mathbf{x}\sim\hat{\mathcal{D}}_{S_i}}(h(\mathbf{x}=0)) \right) \\
&= \max_{i\in[k]} \left( 1 - 2\min_{h\in\mathcal{H}\Delta\mathcal{H}} \left( \frac{1}{2m}\sum_{\mathbf{x}\sim\hat{\mathcal{D}}_T} \mathbb{I}(h(\mathbf{x})=1) + \frac{1}{2m}\sum_{\mathbf{x}\sim\hat{\mathcal{D}}_{S_i}} \mathbb{I}(h(\mathbf{x}=0)) \right) \right)
\end{aligned}
$$

## C.5 Proof of Thm. 4.1

**Theorem 4.1.** Let $\mathcal{D}_T$ and $\{\mathcal{D}_{S_i}\}_{i=1}^k$ be the target distribution and $k$ source distributions over $\mathcal{X}$. Let $\mathcal{H}$ be a hypothesis class where $VC\dim(\mathcal{H}) = d$. If $\hat{\mathcal{D}}_T$ and $\{\hat{\mathcal{D}}_{S_i}\}_{i=1}^k$ are the empirical distributions of $\mathcal{D}_T$ and $\{\mathcal{D}_{S_i}\}_{i=1}^k$ generated with $m$ i.i.d. samples from each domain, then, $\forall \alpha \in \mathbb{R}_+^k, \sum_{i\in[k]} \alpha_i = 1$, for $0 < \delta < 1$, with probability at least $1 - \delta$ (over the choice of samples), we have:

$$
\varepsilon_T(h) \leq \sum_{i\in[k]} \alpha_i \cdot \left( \hat{\varepsilon}_{S_i}(h) + \frac{1}{2}d_{\mathcal{H}\Delta\mathcal{H}}(\hat{\mathcal{D}}_T; \hat{\mathcal{D}}_{S_i}) \right) + O\left( \sqrt{\frac{1}{m}\left( \log\frac{k}{\delta} + d\log\frac{me}{d} \right)} \right) + \lambda_\alpha \quad (7)
$$

where $\lambda_\alpha$ is a constant that only depends on $\mathcal{H}$.

*Proof.* We first extend the definition of $\mathcal{H}$-divergence in the multiple sources setting from Def. 3.1 to the case where a convex combination $\alpha$ is used to combine all the single source divergence measures. Under this new divergence measure, we prove similar concentration results like Thm. 3.1, Thm. 3.2 and Thm. 3.3. By choosing $\alpha$ to be proportional to $\exp(\hat{\varepsilon}_i)$, we obtain the upper bound in Thm. 4.1 using a combination of Jensen's inequality and the arithmetic-geometric mean inequality.

Let $\alpha \in \mathbb{R}^k$ be $\alpha \geq 0$ and $\sum_{i\in[k]} \alpha_i = 1$. Define $d_{\mathcal{H},\alpha}(\mathcal{D}_T; \{\mathcal{D}_{S_i}\}_{i=1}^k)$ as follows:

**Definition C.1.**

$$
d_{\mathcal{H},\alpha}(\mathcal{D}_T; \{\mathcal{D}_{S_i}\}_{i=1}^k) := \sum_{i\in[k]} \alpha_i \cdot d_{\mathcal{H}}(\mathcal{D}_T; \mathcal{D}_{S_i}) = 2\sum_{i\in[k]} \alpha_i \cdot \sup_{A\in\mathcal{A}_{\mathcal{H}}} |\Pr_{D_T}(A) - \Pr_{D_{S_i}}(A)|
$$

It is easy to check that Def. C.1 is a generalization of Def. 3.1 where $\alpha$ is chosen to be a one-hot vector that has value 1 in the source domain with the largest discrepancy. Similarly, define $h_\alpha^*$ and $\lambda_\alpha$ as follows:

$$
h_\alpha^* := \arg\min_{h\in\mathcal{H}} \left( \varepsilon_T(h) + \sum_{i\in[k]} \alpha_i \cdot \varepsilon_{S_i}(h) \right), \quad \lambda_\alpha := \varepsilon_T(h^*) + \sum_{i\in[k]} \alpha_i \cdot \varepsilon_{S_i}(h^*)
$$

Realizing that the proof of Thm. 3.1 only depends on the subadditivity of the max operator, and the fact that convex combination trivially satisfies subadditivity, we can easily show that the following generalization bound holds for all such $\alpha$:

$$
\varepsilon_T(h) \leq \sum_{i\in[k]} \alpha_i \cdot \varepsilon_{S_i}(h) + \lambda_\alpha + \frac{1}{2}d_{\mathcal{H}\Delta\mathcal{H},\alpha}(\mathcal{D}_T; \{\mathcal{D}_{S_i}\}_{i=1}^k) \quad (9)
$$

The next step is to provide finite sample bounds for the first and the third terms of (9). To bound the third term, we simply replace $d_{\mathcal{H}}(\mathcal{D}_T; \{\mathcal{D}_{S_i}\}_{i=1}^k)$ in the proof of Thm. 3.2 to $d_{\mathcal{H},\alpha}(\mathcal{D}_T; \{\mathcal{D}_{S_i}\}_{i=1}^k)$, and use the following inequality:

$$\Pr\left(\sum_{i\in[k]} \alpha_i \cdot \sup_{A\in\mathcal{A}_{\mathcal{H}}} \left| |\Pr_{\mathcal{D}_T}(A) - \Pr_{\mathcal{D}_{S_i}}(A)| - |\Pr_{\hat{\mathcal{D}}_T}(A) - \Pr_{\hat{\mathcal{D}}_{S_i}}(A)| \right| \geq \frac{\epsilon}{2}\right)$$

$$\leq \Pr\left(\exists i\in[k], \exists A\in\mathcal{A}_{\mathcal{H}} : \left| |\Pr_{\mathcal{D}_T}(A) - \Pr_{\mathcal{D}_{S_i}}(A)| - |\Pr_{\hat{\mathcal{D}}_T}(A) - \Pr_{\hat{\mathcal{D}}_{S_i}}(A)| \right| \geq \frac{\epsilon}{2}\right)$$

$$\leq \sum_{i=1}^k \Pr\left(\exists A\in\mathcal{A}_{\mathcal{H}} : \left| |\Pr_{\mathcal{D}_T}(A) - \Pr_{\mathcal{D}_{S_i}}(A)| - |\Pr_{\hat{\mathcal{D}}_T}(A) - \Pr_{\hat{\mathcal{D}}_{S_i}}(A)| \right| \geq \frac{\epsilon}{2}\right)$$

where the first inequality is due to the fact that $\sum_{i\in[k]} \alpha_i t_i \geq \epsilon/2 \implies \exists i\in[k], t_i \geq \epsilon/2$, otherwise $\sum_{i\in[k]} \alpha_i t_i < \epsilon/2$, and the second inequality is a simple union bound. All the other parts of the proof of Thm. 3.2 still hold under $d_{\mathcal{H},\alpha}(\mathcal{D}_T; \{\mathcal{D}_{S_i}\}_{i=1}^k)$, hence we immediately have the following lemma to estimate $d_{\mathcal{H},\alpha}(\mathcal{D}_T; \{\mathcal{D}_{S_i}\}_{i=1}^k)$ using finite samples:

**Lemma C.3.** Let $\mathcal{D}_T$ and $\{\mathcal{D}_{S_i}\}_{i=1}^k$ be the target distribution and $k$ source distributions over $\mathcal{X}$. Let $\mathcal{H}$ be a hypothesis class where $VC\dim(\mathcal{H}) = d$. If $\hat{\mathcal{D}}_T$ and $\{\hat{\mathcal{D}}_{S_i}\}_{i=1}^k$ are the empirical distributions of $\mathcal{D}_T$ and $\{\mathcal{D}_{S_i}\}_{i=1}^k$ generated with $m$ i.i.d. samples from each domain, then for $\epsilon > 0$, we have:

$$\Pr\left(\left|d_{\mathcal{H},\alpha}(\mathcal{D}_T; \{\mathcal{D}_{S_i}\}_{i=1}^k) - d_{\mathcal{H},\alpha}(\hat{\mathcal{D}}_T; \{\hat{\mathcal{D}}_{S_i}\}_{i=1}^k)\right| \geq \epsilon\right) \leq 4k\left(\frac{em}{d}\right)^d \exp\left(-m\epsilon^2/8\right)$$

To bound the first term uniformly for all $h \in \mathcal{H}$, we have:

**Lemma C.4.** Let $\{\mathcal{D}_{S_i}\}_{i=1}^k$ be $k$ source distributions over $\mathcal{X}$. Let $\mathcal{H}$ be a hypothesis class where $VC\dim(\mathcal{H}) = d$. If $\{\hat{\mathcal{D}}_{S_i}\}_{i=1}^k$ are the empirical distributions of $\{\mathcal{D}_{S_i}\}_{i=1}^k$ generated with $m$ i.i.d. samples from each domain, then, for $\epsilon > 0$, we have:

$$\Pr\left(\sup_{h\in\mathcal{H}} \left|\sum_{i\in[k]} \alpha_i \cdot \varepsilon_{S_i}(h) - \sum_{i\in[k]} \alpha_i \cdot \hat{\varepsilon}_{S_i}(h)\right| \geq \epsilon\right) \leq 2k\left(\frac{me}{d}\right)^d \exp(-2m\epsilon^2)$$

The proof of the above lemma is as follows:

$$\Pr\left(\sup_{h\in\mathcal{H}} \left|\sum_{i\in[k]} \alpha_i \cdot \varepsilon_{S_i}(h) - \sum_{i\in[k]} \alpha_i \cdot \hat{\varepsilon}_{S_i}(h)\right| \geq \epsilon\right) \leq \Pr\left(\sup_{h\in\mathcal{H}} \sum_{i\in[k]} \alpha_i \left|\varepsilon_{S_i}(h) - \hat{\varepsilon}_{S_i}(h)\right| \geq \epsilon\right)$$

$$\leq \Pr\left(\sum_{i\in[k]} \alpha_i \cdot \sup_{h\in\mathcal{H}} \left|\varepsilon_{S_i}(h) - \hat{\varepsilon}_{S_i}(h)\right| \geq \epsilon\right)$$

$$\leq \Pr\left(\exists i\in[k] : \sup_{h\in\mathcal{H}} \left|\varepsilon_{S_i}(h) - \hat{\varepsilon}_{S_i}(h)\right| \geq \epsilon\right)$$

$$\leq \sum_{i=1}^k \Pr\left(\sup_{h\in\mathcal{H}} \left|\varepsilon_{S_i}(h) - \hat{\varepsilon}_{S_i}(h)\right| \geq \epsilon\right)$$

$$\leq k \cdot \Pi_{\mathcal{H}}(m) \Pr\left(\left|\varepsilon_{S_i}(h) - \hat{\varepsilon}_{S_i}(h)\right| \geq \epsilon\right)$$

$$\leq k \cdot \Pi_{\mathcal{H}}(m) \cdot 2\exp(-2m\epsilon^2)$$

$$\leq 2k\left(\frac{me}{d}\right)^d \exp(-2m\epsilon^2)$$

The first inequality is due to the triangle inequality of $|\cdot|$, and the second one is because of the subadditivity of the $\sup$ function. The third inequality holds by a contrapositive argument. The fourth one is by the union bound. We apply the VC-inequality to bound the fifth inequality, and Hoeffding's inequality to bound the sixth. Again, the last one is due to Sauer's lemma.

To complete the proof, we simply combine Lemma C.3 and Lemma C.4 into (9), and solve for $\epsilon$. $\blacksquare$

## C.6 PROOF OF THM. 4.2

**Theorem 4.2.** Choose $\alpha_i = \exp(\widehat{\varepsilon}_i(h))/\sum_{j\in[k]}\exp(\widehat{\varepsilon}_j(h))$ with $\widehat{\varepsilon}_i(h) := \widehat{\varepsilon}_{S_i}(h) + \frac{1}{2}d_{\mathcal{H}\Delta\mathcal{H}}(\widehat{\mathcal{D}}_T; \widehat{\mathcal{D}}_{S_i})$, we have

$$\varepsilon_T(h) \leq \log\sum_{i\in[k]}\exp(\widehat{\varepsilon}_i(h)) + O\left(\sqrt{\frac{1}{m}\left(\log\frac{k}{\delta} + d\log\frac{me}{d}\right)}\right) + \lambda^* \tag{8}$$

where $\lambda^*$ is a constant that only depends on $\mathcal{H}$.

*Proof.* Define $\alpha_i = \exp(\widehat{\varepsilon}_i(h))/\sum_{j\in[k]}\exp(\widehat{\varepsilon}_j(h))$ with $\widehat{\varepsilon}_i(h) := \widehat{\varepsilon}_{S_i}(h) + \frac{1}{2}d_{\mathcal{H}\Delta\mathcal{H}}(\widehat{\mathcal{D}}_T; \widehat{\mathcal{D}}_{S_i})$, then the first term of the R.H.S. of (7) can be bounded as:

$$\sum_{i\in[k]}\frac{\exp(\widehat{\varepsilon}_i(h))}{\sum_{j\in[k]}\exp(\widehat{\varepsilon}_j(h))}\cdot\widehat{\varepsilon}_i(h) = \mathbb{E}_\alpha[\widehat{\varepsilon}_i(h)] = \mathbb{E}_\alpha[\log\exp(\widehat{\varepsilon}_i(h))]$$

$$\leq \log\left(\mathbb{E}_\alpha[\exp(\widehat{\varepsilon}_i(h))]\right)$$

$$= \log\left(\frac{\sum_{i\in[k]}\exp^2(\widehat{\varepsilon}_i(h))}{\sum_{i\in[k]}\exp(\widehat{\varepsilon}_i(h))}\right)$$

$$\leq \log\sum_{i\in[k]}\exp(\widehat{\varepsilon}_i(h))$$

where the first inequality is due to the Jensen's inequality, and the second one is based on the fact that $\sum_i a_i^2 \leq (\sum_i a_i)^2$ when $a_i \geq 0, \forall i$. ∎

# D DETAILS ABOUT EXPERIMENTS

In this section, we describe more details about the datasets and the experimental settings. We extensively evaluate the proposed methods on three datasets: 1). We first evaluate our methods on Amazon Reviews dataset (Chen et al., 2012) for sentiment analysis. 2). We evaluate the proposed methods on the digits classification datasets including MNIST (LeCun et al., 1998), MNIST-M (Ganin et al., 2016), SVHN (Netzer et al., 2011), and SynthDigits (Ganin et al., 2016). 3). We further evaluate the proposed methods on the public dataset WebCamT (Zhang et al., 2017) for vehicle counting. It contains 60,000 labeled images from 12 city cameras with different distributions. Due to the substantial difference between these datasets and their corresponding learning tasks, we will introduce more detailed dataset description, network architecture, and training parameters for each dataset respectively in the following subsections.

## D.1 DETAILS ON AMAZON REVIEWS EVALUATION

Amazon reviews dataset includes four domains, each one composed of reviews on a specific kind of product (Books, DVDs, Electronics, and Kitchen appliances). Reviews are encoded as 5000 dimensional feature vectors of unigrams and bigrams. The labels are binary: 0 if the product is ranked up to 3 stars, and 1 if the product is ranked 4 or 5 stars.

We take one product domain as target and the other three as source domains. Each source domain has 2000 labeled examples and the target test set has 3000 to 6000 examples. We implement the Hard-Max and Soft-Max methods according to Alg. 1, based on a basic network with one input layer (5000 units) and three hidden layers (1000, 500, 100 units). The network is trained for 50 epochs with dropout rate 0.7. We compare Hard-Max and Soft-Max with three baselines: *Baseline 1: MLPNet*. It is the basic network of our methods (one input layer and three hidden layers), trained for 50 epochs with dropout rate 0.01. *Baseline 2: Marginalized Stacked Denoising Autoencoders (mSDA)* (Chen et al., 2012). It takes the unlabeled parts of both source and target samples to learn a feature map from input space to a new representation space. As a denoising autoencoder algorithm, it finds a feature representation from which one can (approximately) reconstruct the original features of an example from its noisy counterpart. *Baseline 3: DANN*. We implement DANN based on the algorithm

described in (Ganin et al., 2016) with the same basic network as our methods. Hyper parameters of the proposed and baseline methods are selected by cross validation. Table 5 summarizes the network architecture and some hyper parameters.

Table 5: Network parameters for proposed and baseline methods

| Method | Input layer | Hidden layers | Epochs | Dropout | Domains | Adaptation weight | $\gamma$ |
|---|---|---|---|---|---|---|---|
| **MLPNet** | 5000 | (1000, 500, 100) | 50 | 0.01 | N/A | N/A | N/A |
| **DANN** | 5000 | (1000, 500, 100) | 50 | 0.01 | 1 | 0.01 | N/A |
| **MDAN** | 5000 | (1000, 500, 100) | 50 | 0.7 | 3 | 0.1 | 10 |

### D.2 DETAILS ON DIGIT DATASETS EVALUATION

We evaluate the proposed methods on the digits classification problem. Following the experiments in (Ganin et al., 2016), we combine four popular digits datasets-MNIST, MNIST-M, SVHN, and SynthDigits to build the multi-source domain dataset. MNIST is a handwritten digits database with $60,000$ training examples, and $10,000$ testing examples. The digits have been size-normalized and centered in a $28 \times 28$ image. MNIST-M is generated by blending digits from the original MNIST set over patches randomly extracted from color photos from BSDS500 (Arbelaez et al., 2011; Ganin et al., 2016). It has $59,001$ training images and $9,001$ testing images with $32 \times 32$ resolution. An output sample is produced by taking a patch from a photo and inverting its pixels at positions corresponding to the pixels of a digit. For DA problems, this domain is quite distinct from MNIST, for the background and the strokes are no longer constant. SVHN is a real-world house number dataset with $73,257$ training images and $26,032$ testing images. It can be seen as similar to MNIST, but comes from a significantly harder, unsolved, real world problem. SynthDigits consists of $500;000$ digit images generated by Ganin et al. (2016) from WindowsTM fonts by varying the text, positioning, orientation, background and stroke colors, and the amount of blur. The degrees of variation were chosen to simulate SVHN, but the two datasets are still rather distinct, with the biggest difference being the structured clutter in the background of SVHN images.

We take MNIST-M, SVHN, and MNIST as target domain in turn, and the remaining three as sources. We implement the Hard-Max and Soft-Max versions according to Alg. 1 based on a basic network, as shown in Fig. 5. The baseline methods are also built on the same basic network structure to put them on a equal footing. The network structure and parameters of MDANs are illustrated in Fig. 5. The learning rate is initialized by $0.01$ and adjusted by the first and second order momentum in the training process. The domain adaptation parameter of MDANs is selected by cross validation. In each mini-batch of MDANs training process, we randomly sample the same number of unlabeled target images as the number of the source images.

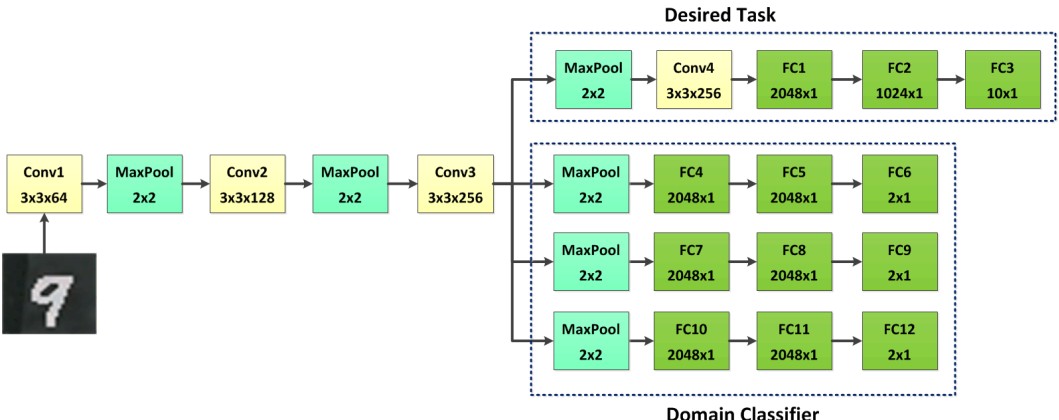

Figure 5: MDANs network architecture for digit classification

### D.3 Details on WebCamT Vehicle Counting

WebCamT is a public dataset for large-scale city camera videos, which have low resolution ($352 \times 240$), low frame rate (1 frame/second), and high occlusion. WebCamT has $60,000$ frames annotated with rich information: bounding box, vehicle type, vehicle orientation, vehicle count, vehicle re-identification, and weather condition. The dataset is divided into training and testing sets, with 42,200 and 17,800 frames, respectively, covering multiple cameras and different weather conditions. WebCamT is an appropriate dataset to evaluate domain adaptation methods, for it covers multiple city cameras and each camera is located in different intersection of the city with different perspectives and scenes. Thus, each camera data has different distribution from others. The dataset is quite challenging and in high demand of domain adaptation solutions, as it has $6,000,000$ unlabeled images from 200 cameras with only $60,000$ labeled images from 12 cameras. The experiments on WebCamT provide an interesting application of our proposed MDANs: when dealing with spatially and temporally large-scale dataset with much variations, it is prohibitively expensive and time-consuming to label large amount of instances covering all the variations. As a result, only a limited portion of the dataset can be annotated, which can not cover all the data domains in the dataset. MDAN provide an effective solution for this kind of application by adapting the deep model from multiple source domains to the unlabeled target domain.

We evaluate the proposed methods on different numbers of source cameras. Each source camera provides 2000 labeled images for training and the test set has 2000 images from the target camera. In each mini-batch, we randomly sample the same number of unlabeled target images as the source images. We implement the Hard-Max and Soft-Max version of MDANs according to Alg. 1, based on the basic vehicle counting network FCN described in (Zhang et al., 2017). Please refer to (Zhang et al., 2017) for detailed network architecture and parameters. The learning rate is initialized by 0.01 and adjusted by the first and second order momentum in the training process. The domain adaptation parameter is selected by cross validation. We compare our method with two baselines: *Baseline 1: FCN*. It is our basic network without domain adaptation as introduced in work (Zhang et al., 2017). *Baseline 2: DANN*. We implement DANN on top of the same basic network following the algorithm introduced in work (Ganin et al., 2016).

## E More Related Work

A number of adaptation approaches have been studied in recent years. From the theoretical aspect, several theoretical results have been derived in the form of upper bounds on the generalization target error by learning from the source data. A keypoint of the theoretical frameworks is estimating the distribution shift between source and target. Kifer et al. (2004) proposed the $\mathcal{H}$-divergence to measure the similarity between two domains and derived a generalization bound on the target domain using empirical error on the source domain and the $\mathcal{H}$-divergence between the source and the target. This idea has later been extended to multisource domain adaptation (Blitzer et al., 2008) and the corresponding generalization bound has been developed as well. Ben-David et al. (2010) provide a generalization bound for domain adaptation on the target risk which generalizes the standard bound on the source risk. This work formalizes a natural intuition of DA: reducing the two distributions while ensuring a low error on the source domain and justifies many DA algorithms. Based on this work, Mansour et al. (2009a) introduce a new divergence measure: discrepancy distance, whose empirical estimate is based on the Rademacher complexity (Koltchinskii, 2001) (rather than the VC-dim). Other theoretical works have also been studied such as (Mansour & Schain, 2012) that derives the generalization bounds on the target error by taking use of the robustness properties introduced in (Xu & Mannor, 2012). See (Cortes et al., 2008; Mansour et al., 2009a;c) for more details.

Following the theoretical developments, many DA algorithms have been proposed, such as instance-based methods (Tsuboi et al., 2009); feature-based methods (Becker et al., 2013); and parameter-based methods (Evgeniou & Pontil, 2004). The general approach for domain adaptation starts from algorithms that focus on linear hypothesis class (Blitzer et al., 2006; Germain et al., 2013; Cortes & Mohri, 2014). The linear assumption can be relaxed and extended to the non-linear setting using the kernel trick, leading to a reweighting scheme that can be efficiently solved via quadratic programming (Huang et al., 2006; Gong et al., 2013). Recently, due to the availability of rich data and powerful computational resources, non-linear representations and hypothesis classes have been increasingly explored (Glorot et al., 2011; Baktashmotlagh et al., 2013; Chen et al., 2012; Ajakan

et al., 2014; Ganin et al., 2016). This line of work focuses on building common and robust feature representations among multiple domains using either supervised neural networks (Glorot et al., 2011), or unsupervised pretraining using denoising auto-encoders (Vincent et al., 2008; 2010).

Recent studies have shown that deep neural networks can learn more transferable features for DA (Glorot et al., 2011; Donahue et al., 2014; Yosinski et al., 2014). Bousmalis et al. (2016) develop domain separation networks to extract image representations that are partitioned into two subspaces: domain private component and cross-domain shared component. The partitioned representation is utilized to reconstruct the images from both domains, improving the DA performance. Reference (Long et al., 2015) enables classifier adaptation by learning the residual function with reference to the target classifier. The main-task of this work is limited to the classification problem. Ganin et al. (2016) propose a domain-adversarial neural network to learn the domain indiscriminate but main-task discriminative features. Although these works generally outperform non-deep learning based methods, they only focus on the single-source-single-target DA problem, and much work is rather empirical design without statistical guarantees. Hoffman et al. (2012) present a domain transform mixture model for multisource DA, which is based on non-deep architectures and is difficult to scale up.

Adversarial training techniques that aim to build feature representations that are indistinguishable between source and target domains have been proposed in the last few years (Ajakan et al., 2014; Ganin et al., 2016). Specifically, one of the central ideas is to use neural networks, which are powerful function approximators, to approximate a distance measure known as the $\mathcal{H}$-divergence between two domains (Kifer et al., 2004; Ben-David et al., 2007; 2010). The overall algorithm can be viewed as a zero-sum two-player game: one network tries to learn feature representations that can fool the other network, whose goal is to distinguish representations generated from the source domain between those generated from the target domain. The goal of the algorithm is to find a Nash-equilibrium of the game, or the stationary point of the min-max saddle point problem. Ideally, at such equilibrium state, feature representations from the source domain will share the same distributions as those from the target domain, and, as a result, better generalization on the target domain can be expected by training models using only labeled instances from the source domain.

