# OpenReview forum: "Multiple Source Domain Adaptation with Adversarial Learning"
_ICLR.cc/2018/Conference — Invite to Workshop Track_

### Official Review · AnonReviewer1 · 2017-11-26
**The theoretical analysis is new and interesting but for me its worst case aspect does not reflect real application of multisource learning and the experimental results are good but lack of comparisons with other multi-source approaches.**

**Rating:** 6
**Confidence:** 3

**Review:**

Quality:
The paper appears to be correct.

Clarity:
The paper is very clear

Originality:
The theoretical contribution extends the seminal work of Ben-David et al., the idea of using adversarial learning is not new, the novelty is mediaum

Significance:
The theoretical analysis is interested but for me limited, the idea of the algorithm is not new but as far as I know the first explicitly presented for multi-source.

Pros:
-new theoretical analysis for multisource problem
-paper clear
-smoothed version is interesting
Cons
-Learning bounds with worst case standpoint is probably not the best analysis for multisource learning
-experimental evaluation limited in the sense that similar algorithms in the literature are not compared
-Extension a bit direct from the seminal work of Ben-David et al.


Summary:
This paper presents a multiple source domain adaptation approach based on adversarial learning.
The setting considered contains multiple source domains with labeled instances and one target domain with unlabeled instances. The authors propose learning bounds in this context that extend seminal work of Ben-David and co-authors(2007,2010) where they essentially consider the max source error and the max divergence between target and source with the presence of empirical estimate.
Then, they propose an adversarial algorithm to optimize this bound, with another version optimizing a smoothed version, following the approach of Ganin et al.(2016).
An experimental evaluation on 3 known tasks is presented.

Comments:
Comments:

-I am not particularly convinced that the proposed theory explains best multi-source learning. In multi-source, you expect that one source may compensate the others when needed for classification of particular instances. The paper considers a kind of worst case by taking the max error over the sources and the max divergence between target and source, but not really representative of what happens for real problems in the sense that you do not take into account how the different sources interact.
The experimental results confirm this aspect actually.
Maybe, the authors could propose a learning bound that correspond to the smoothed version proposed in the paper and that works best.

The Hard version of the algorithm seems here to comply with the bound, while the algorithm that is really interesting is the smoothed version.

-Experimental evaluation is a bit limited, there is no comparison with other (deep learning methods) tackling multi-source scenarios (or equivalent), while I think it is easy to find related approaches :
-E. Tzeng, J. Hoffman, T. Darrell, K. Saenko. Simultaneous Deep Transfer Across Domains and Tasks. ICCV 2015.
-I-H Jhuo, D Liu, D.T. Lee, and S-Fu. Chang. Robust visual domain adaptation with low-rank reconstruction. In IEEE CVPR, 2012.
-Muhammad Ghifary, W. Bastiaan Kleijn, Mengjie Zhang, and David Balduzzi. Domain generalization for object recognition with multi-task autoencoders. In IEEE International Conference on Computer Vision (ICCV), 2015.
-Chuang Gan, Tianbao Yang, and Boqing Gong. Learning attributes equals multi-source domain generalization. In IEEE Conference on Computer Vision and Pattern Recognition (CVPR), 2016.
-R. Gopalan,R. Li,and R. Chellappa. Unsupervised Adaptation Across Domain shifts by generating intermediate data representations. PAMI, 36(11), 2014.

Note also this paper at CVPR'17: about Domain adversarial adaptation.
E. Tzeng, J. Hoffman, K. Saenko, T. Darrell.  Adversarial Discriminative Domain Adaptation, CVPR 2017.


-Nothing is said about the complexity of applying the algorithm on the different datasets (convergence, tuning, ...)
For the smoothed version, it could be interesting to see if the weights w_i associated to each source are related to each (original) source error and see how the sources are complementary.

--
After rebuttal
--
The new results and experimental evaluation have improved the paper. I increased my score.

---

> ### Author Response · Authors · 2017-12-14
> **Response to Reviewer 1**
>
> We thank the reviewer for agreeing that the theoretical analysis is new and interesting. Besides learning bounds with worst case, we also proposed a smoothed version (Equation (5) in Section 4) and prove a new generalization bound where the minimization of the smoothed version corresponds to the minimization of this upper bound (Theorem 4.1 and Theorem 4.2 in Section 4), which provides a theoretical justification for the optimization of Equation (5). As explained in the Common Remarks, instead of considering the worst-case scenario, this new bound is obtained by considering interactions between multiple source domains, i.e., all the source domains contribute to the upper bound (not just the worst one as stated in Thm. 3.4), and the combination weight of each source domain depends exactly on its empirical error and its distance to the target domain.
>
> We would like to point out that this paper is not just a direct extension of the seminal work of Ben-David et al. We provide detailed comparisons with existing work (in Section 3 "Comparison with Existing Bounds"). For the def. 3.1, it’s not easy to see how to use the convexity property of the max function to obtain a proper upper bound. Both sample complexity bounds given in Thm. 3.4 and Thm. 4.1 are optimal in terms of the number of training instances m in each source domain, as it matches the \Omega(sqrt{1/m}) lower bound in the non-realizable binary classification scenario (see Remark under Theorem 4.2). Those theoretical results are of insights and practical impacts, providing an effective way to train DNN on multiple datasets with good guarantee of the performance. The proposed multi-domain adversarial network is new architecture with impressively good performance.
>
> For the experimental evaluation, we extensively evaluate the proposed methods on three real-world datasets: sentiment analysis, digit classification, and vehicle counting, all showing superior adaptation performance over the baselines. We thank the reviewer for suggesting some related work and add three of them as new baselines in the revised version. Please refer to the Common Remarks for more explanation. Experimental results still show that our method achieves the state-of-the-art performance for multi-source domain adaptation.

---

> ### Author Response · Authors · 2018-01-10
> **Thanks for increasing the rating score!**
>
> Thanks a lot for increasing the rating score of our paper. We appreciate your time and response!

---

### Official Review · AnonReviewer3 · 2017-11-27
**A modest but honest extension of single source adversarial network to multi-source domain adaptation**

**Rating:** 6
**Confidence:** 5

**Review:**

The paper builds on the previous work of Ganin et al. (2015, 2016), that introduced a domain adversarial neural network (DANN) for single source domain adaptation. Whereas Ganin et al. (2016) were building directly on the (single source) domain adaptation theorem of Ben-David et al., the authors prove a similar result for the multiple sources case.

This new result appears to be a simple extension of the single source theorem. A similar result to Theorem 3.1 can be obtained by considering the maximum over the k bounds obtained by considering the k pairs source-target one by one, using Theorem 2.1 of Blitzer et al. (2008). In fact, the latter bound might even be tighter, as Theorem 3.1 considers the maximum over the three components of the domain adaptation bound separately (the source error, the  discrepancy and the lambda term). The same observation holds for Theorem 3.4, which is very similar to Theorem 1 of Blitzer et al. (2008).  This made me doubt that the derived theorem is studying multi-source domain adaptation in an optimal way.
That being said, the authors show in their experiments that their multiple sources network (named MDAN), which is based on their theoretical study, generally achieves better accuracy than the best single source DANN algorithm. This succeeds in convincing me that the proposed approach is of interest. At least, these empirical results could be used as non-trivial benchmarks for further development.

Note that the fact that the "smoothed version" of MDAN performs better than the "hard version", while the latter is directly backed by the theory, also suggests that something is not captured by the theorem. The authors suggest that it can be a question of "data-efficiency performance": "We argue that with more training iterations, the performance of Hard-Max can be further improved" (page 8). This appears to me to be the weakest claim of the paper, since it is not backed by an empirical or a theoretical study.

Pros:
- Tackle an important problem that is not studied as it deserves.
- Based on a theoretical study of the multi-source domain adaptation problem.
- The empirical study is exhaustive enough to show that the proposed algorithm actually works.
- May be used as a benchmark for further multi-source domain adaptation research.

Cons:
- The soft-max version of the algorithm - obtaining the best empirical study - is not backed by the theory.
- It is not obvious that the theoretical study and the proposed algorithm is actually the right thing to do.

Minor comment:
- Section 5: It seems that the benchmarks named "sDANN" and "cDANN" in 5.1 are the same as "best-Single-DANN" and "Combine-DANN" in 5.2. If I am right, the nomenclature must be uniformized.

---

> ### Author Response · Authors · 2017-12-14
> **Response to Reviewer 3**
>
> We would like to thank reviewer 3 for providing thoughtful comments. Please see the revised paper (Theorem 4.1 and Theorem 4.2 in Section 4) for a new bound we proved to justify the smoothed version.
>
> Both Thm. 3.4 and Thm. 4.1 are optimal in terms of the number of training instances m in each source domain, as it matches the \Omega(sqrt{1/m}) lower bound in the non-realizable binary classification scenario. By using different distance measure for distributions, one might get other kinds of bounds that reflect the underlying distance measure (Mansour et al. 2009 a, b, c), but in general those bounds are incomparable to ours, and depending on the concrete settings, one might be tighter than the other.
>
> We would like to point out that the simple strategy by applying the single-source-single-target bound k times cannot be used to derive a bound as we achieved in Thm. 3.4 for the following reasons: the \lambda (error achieved by the optimal hypothesis on S and T) defined in Blitzer 2008 depends on both S and T, hence when combining the k bounds, there does not necessarily exist a single optimal hypothesis h^* that makes this bound hold for all k pairs. Second, in order to use Thm. 1 in Blitzer 2008 to achieve the same result, because of the union bound, one will have to incur an additional square root of log(k) term. On the other hand, this combination technique can indeed be used to show that the asymptotic dependency of the upper bound on m is O(\sqrt{1/m}).
>
> We have changed the nomenclature so that they are consistent in both experiments.

---

> > ### Comment · AnonReviewer3 · 2018-01-12
> > **The authors improved their paper.**
> >
> > I warmly welcome the theoretical study of the smoothed version of the algorithm.
> >
> > However, I maintain my score since I'm still skeptical about the advantage of Theorem 3.4 compared to the maximum over the k single-source bounds. Specifically:
> >
> > 1 - The authors argued that "in order to use Thm. 1 in Blitzer 2008 to achieve the same result, because of the union bound, one will have to incur an additional square root of log(k) term". This is right, but Theorem 3.4 also contains a square root of log(k).
> >
> > 2 - The authors argued that "the \lambda (error achieved by the optimal hypothesis on S and T) defined in Blitzer 2008 depends on both S and T, hence when combining the k bounds, there does not necessarily exist a single optimal hypothesis h^* that makes this bound hold for all k pairs". But, if one takes the maximum over the k bounds (using the union bound as discussed above), he will consider a single pair S and T, which will be valid. In fact, the "multi source" lambda defined in the paper also considers a single pair, given by the minimum of the maximum individual target+source risks (see bottom of page 3). The difference is that the alternative approach will amount at taking the maximum of the minimum individual source risks (which might even give a tighter bound).

---

> > > ### Author Response · Authors · 2018-01-13
> > > **Advantage of our bound**
> > >
> > > We would like to thank Reviewer 3 again for the insightful thoughts and comments. We appreciate your positive feedback for the theoretical study of the smoothed version of the algorithm. We agree with the reviewer that using the single-source-single-target bound k times with union bound can lead to an upper bound that has the same asymptotic order in terms of both m and k. As the reviewer has pointed out, due to the sub-additivity of the max function, this bound is actually tighter. In fact, using the minimax inequality, we can precisely bound the relation of these two lambdas as follows:
> > >
> > > R3's lambda = max_{i \in [k]} min_{h} eps_T(h) + eps_{S_i}(h) <= min_{h} max_{i \in [k]} eps_T(h) + eps_{S_i}(h) = Our lambda
> > >
> > > However, our bound has the advantage that it nicely decouples all the four terms in Thm. 3.4 so that once the dataset and the hypothesis class have been fixed, minimizing the upper bound amounts to minimizing the first two terms. Besides providing an intuitive explanation, the minimization of our upper bound can directly lead to practical learning algorithms (the hard/soft versions) that can be implemented and used. On the other hand, as a comparison, although the alternative upper bound is slightly tighter, it does not admit practical instantiation because minimizing this upper bound requires us to:
> > >
> > > 1.  Compute all the k single-source-single-target bounds.
> > >
> > > 2.  Choose the maximum one to minimize.
> > >
> > > The first step cannot be implemented as it depends on unknown quantity, i.e., the lambda_i for each source domain. In fact, for most interesting hypothesis class, the estimation of lambda_i itself is computationally intractable. Hence, given that both bounds share the same asymptotic complexity and our bound can lead to practical learning algorithms, we still choose to use the current bound.
> > >
> > > We've updated the paper to add this discussion in the remark under Thm 3.4 and acknowledge the Reviewer’s comments.

---

> > > > ### Comment · AnonReviewer3 · 2018-01-13
> > > > **An honest reply**
> > > >
> > > > I thank the authors for their responsiveness. It seems that we reach a common ground. The authors added a comment about the possibility to combine k single-source bounds to obtain a possibly tighter bound (I appreciate the honesty). Intuitively, the fact that we loosen the bound to obtain a more desirable trade-off still makes me believe that it must exist another theoretical analysis of multi-source domain adaptation.
> > > >
> > > > However, the paper experiments show that MDAN achieves generally better empirical result than the best single DANN, which suggests that the authors’ analysis captures something meaningful about the multi-source problem. This is why I consider that the paper contribution is worthy.

---

### Official Review · AnonReviewer4 · 2017-12-03
**A new generalization bound and an extension of using adversarial learning for multiple source domain adaptation**

**Rating:** 6
**Confidence:** 5

**Review:**

The generalization bounds proposed in this paper is an extension of Blitzer et al. 2007. The previous bounds was proposed for single domain single target setting, and this paper extends it to multiple source domain setting.

The proposed bound is presented in Theorem 3.4, showing some interesting observations, such as the performance on the target domain depends on the worst empirical error among multiple source domains.  The proposed bound reduces to Blitzer et al. 2007’s when there is only single source domain.

Pros
+ The proposed bound is of some interest.
+ The bound leads to an efficient learning strategy using adversarial neural networks.

Cons:
- My major concern is that the baselines evaluated in the experiments are quite limited. There are other publications working on the multi-source-domain setting, which were not mentioned/compared in the submission.

---

> ### Author Response · Authors · 2017-12-14
> **Response to Reviewer 4**
>
> We would like to thank the reviewer for providing accurate comments. We have incorporated more comparisons with another three related works for multisource domain adaptation in the experiments. Please check the revised paper (Section 5.2) and the Common Remarks for more details.

---

### Official Review · AnonReviewer2 · 2017-12-04
**An extension of domain adaptation using adversarial learning to multiple source domains.**

**Rating:** 6
**Confidence:** 4

**Review:**

This work presents a bound to learn from multiple source domains for domain adaptation using adversarial learning. This is a simple extension to the previous work based on a single source domain. The adversarial learning aspect is not new.

The proposed method (MDAN) was evaluated on 3 known data sets. Overall, the improvements from using MDAN were consistent and promising.

The bound used in the paper accounts for the worst case scenario, which may not be a tight bound when some of the source domains are very different from the target domain. Therefore, it does not completely address the problem of learning from multiple source domains. The fact that soft-max performs better than hard-max suggest that some form of domain selection or weighting might lead to a better solution. The empirical results in the third experiment (Table 4) also suggest that the proposed solution does not generalize well to domains that are less similar to the target domain.

Some minor comments:
- Section 2: "h disagrees with h" -> "h disagrees with f".
- Theorem 3.1: move the \lambda term to the end to be consistent with equation 1.
- Last line of Section 3: "losses functions" -> "loss functions".
- Table 1 and 2: shorthands sDANN, cDANN, H-Max and S-Max are used here are not consistent with those used in subsequence experiments.  It's good to be consistent.
- In Section 5.2, it was conjectured that the poorer performance of MDAN on SVHN is due to its dissimilarity to the other domains. However, given that the best-single results are close to the target only results, SVHN should be similar to one or more of the source domains. MDAN is probably hurt by the worst case bound.
- In Table 4, the DANN performance for S=6 and T=A is off compared to the rest. Any idea?

---

> ### Author Response · Authors · 2017-12-14
> **Response to Reviewer 2**
>
> Q: “This is a simple extension to the previous work based on a single source domain. The adversarial learning aspect is not new. The bound used in the paper accounts for the worst case scenario, which may not be a tight bound when some of the source domains are very different from the target domain. Therefore, it does not completely address the problem of learning from multiple source domains.”
>
> Thanks for all the suggestions. We would like to take the chance to explain that our theoretical results and algorithms are novel and nontrivial. To our best knowledge, there is no existing work showing the similar theoretical results as ours. Besides, we provide detailed comparisons with existing work (in Section 3 "Comparison with Existing Bounds"). The def. 3.1 is our novel extension to multisource domains, and it’s not easy to see how to use the convexity property of the max function to obtain a proper upper bound. We also prove a new generalization bound where the minimization of the smoothed version corresponds to the minimization of this upper bound, which provides a theoretical justification for the optimization of (5). We precisely state this theorem in Theorem 4.1 and Theorem 4.2. Please see the revised version of the paper about the new upper bound we proved for the smoothed version
>
> As explained in the Common Remarks, instead of considering the worst-case scenario, this new bound is obtained by considering interactions between multiple source domains, i.e., all the source domains contribute to the upper bound (not just the worst one as stated in Thm. 3.4), and the combination weight of each source domain depends exactly on its empirical error and its distance to the target domain.
>
> Those theoretical results are of insights and practical impacts, providing an effective way to train DNN on multiple datasets with good guarantee of the performance. Both sample complexity bounds given in Thm. 3.4 and Thm. 4.1 are optimal in terms of the number of training instances m in each source domain, as it matches the \Omega(sqrt{1/m}) lower bound in the non-realizable binary classification scenario (see Remark under Thm 4.2). The proposed multi-domain adversarial network is new architecture with impressively good performance.
>
>
> Some minor comments:
> Thanks for the detailed review. We have incorporated the minor comments 1~4 in the revised version of the paper.
>
> Q: In Section 5.2, it was conjectured that the poorer performance of MDAN on SVHN is due to its dissimilarity to the other domains. However, given that the best-single results are close to the target only results, SVHN should be similar to one or more of the source domains. MDAN is probably hurt by the worst case bound.
>
> Though the “Hard-Max” has less accuracy, we would like to point out that the smoothed version (“Soft-Max” in table 3) still achieves better performance than the best-Single-Source. Directly applying DANN to the combined source results in even more degraded accuracy compared to “Hard-Max” and “Soft-Max” of MDAN (0.776 v.s. 0.802 & 0.816).
>
> Q: In Table 4, the DANN performance for S=6 and T=A is off compared to the rest. Any idea?
>
> The bad performance of DANN for S=6 and T=A proves our conjecture that directly applying DANN to the combined source leads to suboptimal solutions. We rank the source cameras by their proxy A-distance from the target camera and add them into the source of the experiments one by one. When S=6, the newly added camera is already quite different from the target camera. Without a good mechanism designed for multi-source domain adaptation, directly training DANN with such source data results in lower accuracy. This phenomenon further verifies the necessity of our proposed methods.

---

### Author Response · Authors · 2017-12-14
**Common Remarks (new version of the paper uploaded)**

We thank all the reviewers for the time devoted to provide thoughtful comments and suggestions. We have uploaded a new paper version, in which, following the suggestions of the reviewers, we prove a new generalization bound of the smoothed version and compare with more baselines in the experiments. We attempt to answer the questions  of the reviewers below:

As suggested by reviewers, in the revised version we also prove a new generalization bound where the minimization of the smoothed version corresponds to the minimization of this upper bound, which provides a theoretical justification for the optimization of (5). We precisely state this theorem in Theorem 4.1 and Theorem 4.2 in the revised version, with detailed proof shown in Appendix C.5 and C.6. As a high-level summary, instead of considering the worst-case scenario, this new bound is obtained by considering interactions between multiple source domains, i.e., all the source domains contribute to the upper bound (not just the worst one as stated in Thm. 3.4), and the combination weight of each source domain depends exactly on its empirical error and its distance to the target domain. One can also see that up to constant that does not depend on the training errors of multiple domains, the new upper bound given by Thm. 4.1 is tighter than that of Thm. 3.4. On the other hand, both sample complexity bounds given in Thm. 3.4 and Thm. 4.1 are optimal in terms of the number of training instances m in each source domain, as it matches the \Omega(sqrt{1/m}) lower bound in the non-realizable binary classification scenario (see Remark under Thm 4.2).

We extensively evaluate the proposed methods on three real-world datasets: sentiment analysis, digit classification, and vehicle counting, all showing superior adaptation performance over the baselines. We thank the reviewers to agree with the consistent and promising improvements. We also want to thank Reviewer 1 and Reviewer 4 for pointing out other methods tackling domain adaptation problems, especially for the computer vision problems. While our primary goal is not to achieve state-of-the-art results on specific datasets, we are happy to discuss them in the related work. Besides, we also add more comparisons with these methods in the revised version (Section 5.2). Among the related approaches suggested by Reviewer 1, we found three papers provide codes (Tzeng et al., ICCV 2015; Ghifary et al., ICCV 2015; Tzeng et al., CVPR 2017). We add comparisons and analysis with work (Ghifary et al., ICCV 2015; Tzeng et al., CVPR 2017) in Section 5.2 of the revised version. As the work (Tzeng et al., ICCV 2015) is developed for supervised or semi-supervised domain adaptation (requires some labels for the target domain), while our work is on unsupervised domain adaptation (no label for the target domain)), we didn’t compare with this paper. We also review and add more comparisons with the multi-source-domain adaptation method (Zhang et al. 2015) in Section 5.2 of the revised version as Reviewer 4 suggested. Experimental results still show that our method achieves superior performance for multi-source domain adaptation.

In addition to the above responses, we reply to each reviewer individually for some specific comments.

---

### Decision · Program_Chairs · 2018-01-29
**ICLR 2018 Conference Acceptance Decision**

**Decision:**

Invite to Workshop Track

**Comment:**

Pros
-- Lays out bounds for multi-domain adaptation based on earlier work on a single source-target domain pair.
-- Shows gains over choosing the best source domain for a target domain, or naively combining domains.

Cons
-- The reviewers agree that the extensions are relatively straightforward extensions to single source-target pair.
-- Hard-max doesn’t consider the partial contribution of multiple source domains, and considers the worst-case scenario.
-- Soft-max addresses some of these issues; the authors provide reasonable justification for the algorithm but it’s not clear that the specific choice of \alphas leads to the tightest bound.

The reviewers noted that the authors significantly improved the paper during the revision process. The AC feels that the presented techniques would be of interest to the community and would help lead discussions towards theoretically optimal ways to do domain adaptation given multiple domains. The authors are therefore encouraged to submit to the workshop track.